# Semi-crystalline and amorphous materials via multi-temperature 3D printing from one formulation

Michael Göschl [1,4], Dominik Laa [2,4], Thomas Koch [2], Evan Constable [3], Xin Liu [3], Andrei Pimenov[3], Jürgen Stampfl [2], Robert Liska[1] & Katharina Ehrmann [1] ✉

Multi-material 3D printing concerns the use of two or more 3D printable materials within a single printed part. The result is a composite that benefits from the combined properties of the individual 3D printed materials. Typically, a distinct differentiation between material properties can only be achieved using multiple feedstocks and advanced engineering solutions. In this work, we create multi-material 3D printed photopolymer parts from a single monomer mixture through simple adjustments in printing temperature and light intensity. We achieve this by employing a liquid crystalline (LC) monomer that forms a highly stable LC phase in conjunction with a trifunctional thiol crosslinker. A drastic change in mechanical and optical properties was achieved depending on the presence of an LC phase during polymerization. The proof of principle from bulk experiments could be translated fully into 3D printing, achieving pixel-to-pixel resolution of the material properties solely guided by changing the printing parameters temperature and light intensity. The versatility of produced multi-material composite parts is demonstrated in shape memory applications and methods for chemical data storage and encryption.

3D printing has advanced considerably since its initial use as a proto-typing method. With the advent of material diversity, 3D printing has become more widely accepted as a true alternative method to conventional manufacturing. However, in contrast to fused deposition modeling, stereolithography-based 3D printing still faces the problem that it is inherently designed to manufacture objects out of one material feed. This implies that the entire object consists of the same material. Looking at functional devices, however, e.g., for data storage or soft robotics, a drastic variation of material properties within one device is required to achieve functionality. Therefore, multi-material printing within one object could be highly beneficial to advance 3D printing from prototyping to device fabrication. It would further set the manufacturing technique apart from other types of polymer

processing, which essentially can also only form one type of material per production step (e.g., molding).

In vat photopolymerization, e.g., digital light processing (DLP) printing, a liquid resin mixture consisting of monomers and initiator is filled into the vat and thus determines the final polymer network properties and bulk material properties[1]. Exchanging the material feed, i.e., the vat, offers the ability to print very different material properties within one object. However, this technique is time-intensive and challenging for precise spatio-temporal control of the process[2]. In addition, vat exchanges bear the problem of contaminating the different, sticky resins with each other. To avoid contamination, an additional, time-consuming cleaning step must be used after every exchange[2,3]. Nonetheless, entirely cross-contamination-free 3D

[1]Institute of Applied Synthetic Chemistry, Technische Universität Wien, Getreidemarkt 9/163, 1060 Vienna, Austria. [2]Institute of Materials Science and Technology, Technische Universität Wien, Getreidemarkt 9/308, 1060 Vienna, Austria. [3]Institute of Solid State Physics, Technische Universität Wien, Wiedner Hauptstraße 8, 1040 Vienna, Austria. [4]These authors contributed equally: Michael Göschl, Dominik Laa. ✉e-mail: katharina.ehrmann@tuwien.ac.at

printing via this method could not be implemented so far. Instead of exchanging the material feed, its composition can also be changed during printing (gradient printing)[4]. This allows for some more freedom in the types of properties differentiated. For example, varying degrees of crystallinity could be obtained through such an approach[5]. However, this limits the multi-material property differentiation to a property gradient along the printing direction. Additional difficulties may arise in homogenizing the gradually changing resin composition of the typically rather viscous resins during the printing process.

Therefore, more recent multi-material vat photopolymerization strategies have focused on the change of material properties through the change in irradiation conditions, light intensity and light color[2]. In greyscale printing (in analogy to a black-and-white printer), the intensity of incident light is varied to alter the conversion of the resin into the polymer network and thereby alter its thermomechanical behavior. Hard and soft sections within one object have been fabricated from (meth)acrylate resins at high and low irradiation intensities, respectively, by changing the conversion[6–8]. However, while significant adjustments in curing parameters and therefore monomer conversion can induce moderate property changes, the fundamental type of material (stiff vs. soft, tough vs. brittle) is determined by the type and composition of the building blocks[9,10]. In addition, post-curing may increase the conversion of unreacted monomer, which could not be washed out of a bulk object, in undercured areas. Hence, the property differentiation decreases. Another property differentiation method affects the color of 3D printed samples via greyscale printing, and is based on the oxidation state of a dye as an additive: At high radical concentrations due to high irradiation intensities, the dye is oxidized during printing, opposite non-oxidized dye at low irradiation intensity[11]. Similarly, pH changes have been exploited to change the color of additives in the printing formulation[12]. Again, post-curing could hamper the color differentiation in the long term. Besides free radical photopolymerization, light-triggered cycloaddition reactions such as [2 + 2] dimerization reactions of chromophores pendant to prepolymer chains were utilized as crosslinking reactions in greyscale printing. This led to high-resolution stiffness tuning without decreasing the property differentiation via post-curing[13]. This effect is based on the drastically decreased probability of bond formation or dissociation post-printing because the unreacted groups are immobilized on prepolymers in a rigid matrix, making dimerization in the printed bulk materials highly unlikely. In a variation of this approach, a different type of photocycloaddition dimerization was utilized as a crosslinking reaction for prepolymers. Herein, cycloaddition adducts are produced under green light that are unstable if the forward reaction is not continuously triggered by green light[14]. The formed crosslinked materials thus degrade in the absence of light. Tuning the crosslinking density via greyscale printing tremendously influences the likelihood of cycloadduct reversion and hence the ability of the materials to degrade in darkness: Above a certain crosslinking threshold at high laser energies, entirely undegradable objects were produced.

In contrast to greyscale printing approaches, multi-color printing, or multi-wavelength printing (in analogy to a color printer), refers to 3D printing strategies where different colors (for 3D printing with LED light sources utilizing digital light processing) or wavelengths (for laser-stereolithography) are utilized to address (semi-)orthogonal polymerization reactions during the printing process[15–18]. Most frequently, printing of hard and soft material sections in one object is reported for multi-color printing[19]. This technique relies on semi-orthogonal photopolymerization reactions, where a radical and a cationic photopolymerization reaction can be triggered in two different wavelength regimes (ultraviolet (UV) and visible (vis) light), thereby printing different polymer networks when irradiated with UV or vis light. Importantly, these reactions are not mutually exclusive because UV light typically also triggers the reaction intended for the visible light range, hence the term semiorthogonal. This combination allows for selective vis-induced radical polymerization of a soft (meth) acrylate network in the presence of a UV-active and vis-inactive photoacid generator for cationic photopolymerization. In the UV-region, however, both photoinitiators trigger network formation and a much stiffer, interpenetrating network is created[20,21]. Further studies have added post-modification steps to this approach to vary the dangling chain ends in the soft network and thereby material properties[22]. Another recent report also demonstrates degradable/non-degradable material properties from one resin. Herein, as already introduced previously, two chromophore-based prepolymers are mixed and crosslinked via light-triggered cycloaddition dimerization reactions under conditions that allow fully orthogonal deposition of either polymer network onto the printing platform without the other[23]. Since only one of the two crosslinking reactions is reversible under UV light, this material can be degraded selectively while the second material remains unobstructed.

Most greyscale and multi-color/wavelength printing approaches limit the property differentiation to stiff/soft[16–22], although the differentiation of some other property combinations has been investigated more recently as described above. In addition, there is one limitation common to all these approaches: The final object will contain uncured monomers or prepolymers, which, depending on the crosslinking density, cannot be washed out during the post-processing step. This is particularly problematic because the latent photoinitiators are still in the network and may cure leftover monomers over the lifetime of an object, thereby diminishing the intended property differentiations. In case of (meth)acrylate formulations, even autoinitiation may be sufficient for this process. In other cases, the monomers may slowly diffuse out of the object, causing issues like cavities, sticky surfaces and the release of potentially hazardous monomers.

Therefore, we suggest a paradigm change in multi-material printing by rethinking the approach of obtaining varying material properties from a selection of monomers. Instead of ascribing different material properties to different monomers, we envision "switchable monomers", which change their functionality based on the printing parameters, thereby affecting the obtained material properties.

Herein, we propose the printing temperature as a parameter to alter the material properties mid-printing. We suggest our recently established liquid crystalline thiol-ene monomer platform[24] as a means to affect crystallinity of resulting polymer networks via the printing temperature instead of previously utilized property tuning via alignment through rubbed polymer surfaces, stretching of prepolymers, or external fields[25,26]. This goes beyond previously reported multi-property printing of stiff vs. soft materials, as we additionally focus on optical property differentiation. While 3D printing at moderately high temperature (80 °C) enables trapping of the liquid crystalline state of the formulation in the network, rendering it stiff and opaque, printing at higher temperature (>95 °C) allows for curing of largely amorphous polymer networks from the formulation's isotropic state. Based on this approach, we optimize the property differentiation through finetuning the irradiation protocol (intensity, duration) to highly precise multi-material property printing, with pixel-to-pixel precision property switching within one printing layer, demonstrating first suggestions for application as smart materials for shape memory, chemical data storage and encryption.

## Results and discussion
### Property differentiation studies
Lithography-based 3D printing typically requires fast gelation of a polymer formulation, which is achieved through network formation. Since polymer networks traditionally exhibit amorphous and inhomogeneous microstructures, however, it is particularly difficult to

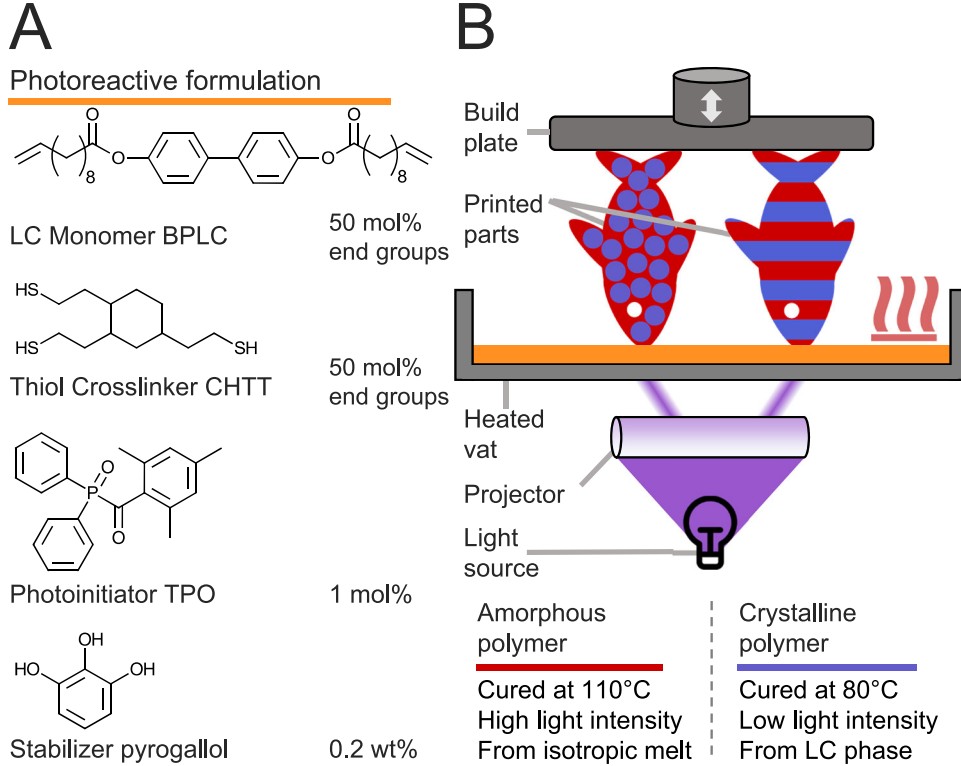

**Fig. 1 | Concept of multi-material 3D printing from a single formulation.**
**A** Photoreactive formulation containing difunctional liquid crystalline terminal alkene monomer BPLC (50 mol% of reactive end groups in the formulation), trifunctional crosslinker CHTT (50 mol% of reactive end groups in the formulation), photoinitiator TPO (1 mol% relative to each type of reactive end group), and stabilizer pyrogallol (0.2 wt% relative to total formulation weight). **B** A heated DLP (digital light processing) 3D printer is programmed to irradiate certain areas of each printed layer at a higher temperature and irradiation intensity, and other areas at a lower intensity and temperature. A photoreactive formulation consisting of the biphenyl-based liquid crystalline monomer BPLC and trithiol crosslinker CHTT, along with a photoinitiator and stabilizer, is employed. At 80 °C, the formulation is present in the liquid crystalline phase, which causes crystalline domains to be incorporated into the polymer network, leading to an opaque appearance and enhanced strength and stiffness (depicted in blue). At 110 °C, the formulation is present as an isotropic molten phase, which, upon polymerization, turns into a transparent, soft, amorphous material (depicted in red). Herein, the decisive parameter in adjusting the crystallinity is the temperature before and during the irradiation process. Low light intensity is utilized when curing the crystalline parts to ensure that the polymerizing formulation does not exceed the LC range through heat of polymerization.

influence the polymer properties via the materials' microstructural architecture[27].

We have recently demonstrated 3D printing of photoreactive liquid crystalline formulations, which contain high amounts of the evenly distributed LC motifs in the main chain of the polymer network by polymerizing a liquid crystalline diene with a trithiol via thiol-ene photopolymerization[24]. This has facilitated uniform and irrevocable formation of crystallinity throughout the material. However, printing from the isotropic phase was not possible for two reasons. First, there was a wide temperature range where a partial LC phase was formed, rendering it impossible to select printing parameters which achieve a clean transition between amorphous and semi-crystalline polymer networks as required for multi-material printing. Secondly, liquid crystallinity persisted until 130 °C, which is too high for the current 3D printing setup. Thus, we tested a different monomer with a lower melting point, looking for a sharper LC-isotropic transition in the formulation.

In this work, another liquid crystalline ene-monomer (BPLC), which exhibits a highly ordered smectic X phase, was synthesized. This monomer is solid at room temperature, transitions into its liquid crystalline state at 77 °C, and its isotropic liquid state at 111 °C[28]. In contrast to our previously reported monomer[24], BPLC was thus expected to be easily processable within the printing window of hot lithography in both its liquid crystalline and isotropic liquid states.

BPLC was then paired with the trifunctional thiol CHTT in a photoreactive formulation (Fig. 1A). CHTT was chosen as it had previously led to the most stable liquid crystalline phases of the formulation and to the highest strength and crystallinity of resulting materials compared to formulations using other trithiols in combination with a liquid crystalline ene[24]. For photoinitiation, TPO was chosen due to its wide availability, a wavelength absorbance range that matches well with our available light sources, and its high reactivity[29]. To stabilize the formulation even at high temperatures, pyrogallol was utilized as it is a common and effective stabilizer in thiol-ene formulations[30]. As complex 3D printing experiments were planned, which necessitate good stability at high temperatures over extended periods of time, a higher stabilizer content of 0.2 wt% was used, compared to the 0.05 wt% utilized in our previous work. Therefore, a higher photoinitiator concentration of 1 mol% had to be employed to achieve sufficient reactivity. A stability test of the formulation resulted in no viscosity change for five hours at a storage temperature of 110 °C, which is sufficient for a variety of 3D printing tests (Supplementary Note 1). However, for even longer printing experiments, further adjustment of the stabilizer content may be necessary.

Polarized optical microscope analysis of the BPLC-based formulation revealed an exceptionally stable LC phase ranging from -79 to 109 °C (Supplementary Note 2), which is present at nearly identical temperatures found for the pure monomer (77 to 111 °C, Supplementary Fig. 6). Compared to previously reported monomers[24], the present formulation exhibited a much sharper transition between its fully LC and fully isotropic phases than any formulation tested in our previous work (LC-isotropic transition taking place within less than 5 °C as

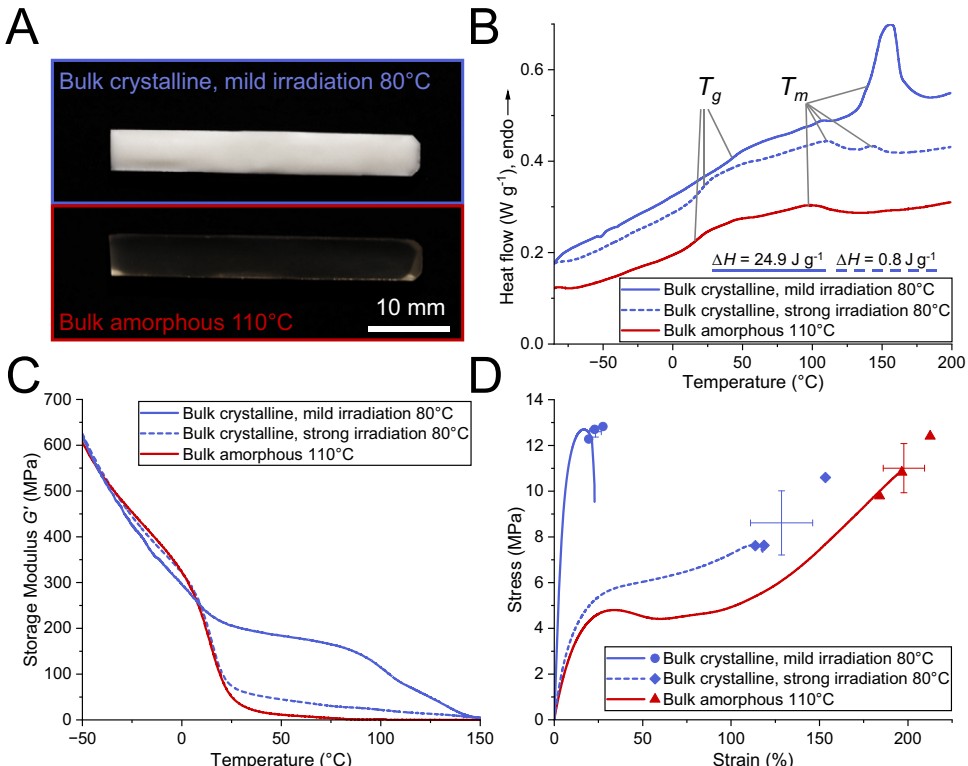

**Fig. 2 | Demonstration of highly variable material properties of bulk cured specimens depending on curing conditions.** Solid blue line: curing temperature 80 °C, mild irradiation (300 s at 1.5 mW cm$^{-2}$ followed by 180 s at 290 mW cm$^{-2}$). Dashed blue line: curing temperature 80 °C, strong irradiation (180 s at 290 mW cm$^{-2}$). Red line: samples cured at 110 °C at strong irradiation. **A** Images of bulk cured specimens highlighting differences in sample transparency depending on curing conditions. Images were recorded at room temperature. **B** Differential scanning calorimetry of bulk cured polymers (DSC) with melt enthalpies ΔH. **C** Thermomechanical properties measured using dynamic mechanical analysis (DMA). **D** Selected tensile test curves with stress and strain at break from all tested specimens. The error bars display the mean ± SD, $n = 3$.

opposed to a nearly 20 °C wide partial liquid crystalline phase in previous formulations), enabling it to be polymerized from either of these distinct phases with a simple adjustment of the printing temperature by only a few degrees. We postulate that this stable LC phase with its sharp LC-isotropic transition allows for much greater control over the resulting polymer's crystallinity by choosing the curing temperature either within the LC or within the isotropic phase.

Indeed, during initial tests, irradiation of the molten formulation at 80 °C resulted in opaque, stiff polymer specimens, indicating significant crystallization. Using real-time photorheology to study the kinetics of the curing process, we confirm that polymer crystallinity forms immediately upon polymerization, as there is no delayed increase in stiffness beyond the expected development seen through polymer network formation (Supplementary Note 3). Moreover, by raising the temperature to 110 °C before irradiation and thus polymerizing from the isotropic phase, we succeeded in producing transparent, amorphous specimens. This property-switchable behavior gave rise to the development of the concept of crystallinity-based multi-material printing from a single formulation (Fig. 1B). By dynamically changing the temperature during the 3D printing process, crystallization can be induced or avoided deliberately. Further adjustment of the parameters to include a low irradiation intensity over a longer irradiation period, as opposed to short intense irradiation, helps reduce temperature spikes through polymerization heat, keeping the formulation within the LC phase. By applying this principle within a single print layer, full 3D control of mechanical and optical properties in a printed part could be achieved.

Along with the evident difference in transparency with different polymerization conditions (Fig. 2A, curing conditions for all polymer samples: Supplementary Table 2), the variability of crystallinity is highlighted by dynamic scanning calorimetry (DSC) measurements, which indicate endo- and exothermal phase transitions in materials upon controlled heating (Fig. 2B and Supplementary Note 4). The softer, amorphous network cured at 110 °C is characterized by a baseline shift associated with the glass transition temperature (T$_g$) and a broad, flat peak around 100 °C that could signify traces of crystallinity[31,32].

For the samples cured at 80 °C, we found that crystallinity could be maximized by employing a mild two-step irradiation procedure, in which low-intensity irradiation (300 s at 1.5 mW cm$^{-2}$) slowly initiates polymer network formation, followed by a second, high-intensity irradiation step (180 s at 290 mW cm$^{-2}$) that finalizes the curing process. In this case, a strong melting peak T$_m$ occurs around 150 °C in the DSC graph (Fig. 2B). This melting point exhibits a very high absolute melt enthalpy of 24.9 J g$^{-1}$, indicating a high degree of crystallinity in comparison with literature[33,34]. If the mild first irradiation step is avoided and the strong final irradiation step is employed immediately (Fig. 2B), the melting peak around 150 °C is severely diminished (melt enthalpies ΔH differing by a factor of 30), and the DSC curve more closely resembles the amorphous specimen. Macroscopically, the sample is still opaque. Along with the T$_g$, a broad and flat melting peak around 100 °C occurs in all samples, which could indicate smaller crystallite sizes or crystallization associated with a different alignment of the polymer chains. These traces likely occur due to small-scale alignment of oligomeric thiol-ene prepolymers that cannot arrange into larger crystalline domains.

The benefit of the initial low-intensity irradiation step on enhancing crystallinity may be twofold: Firstly, the avoidance of heating the sample excessively during polymerization (through heat of polymerization and the light source itself) ensures that the liquid

crystalline phase is maintained, from which polymer crystallinity is formed. Secondly, it is possible that with high-intensity irradiation, the crosslinking process takes place faster than crystallization, immobilizing the polymer network before the polymer chains can align.

The materials' differences in crystallinities result in a fundamental difference in mechanical properties as seen during dynamic mechanical analysis (DMA), where the storage and loss modulus of a material are determined as a function of temperature (Fig. 2C and Supplementary Fig. 10). Herein, the crystalline network's storage modulus is not significantly impacted by the glass transition around 25 °C. Instead, the storage modulus, which serves as a strong indicator for a polymer's stiffness, stays relatively constant and starts to diminish only at 100 °C.

Finally, we reveal the material's tunability using tensile testing (Fig. 2D). While the highly crystalline specimens exhibit high stiffness and an ultimate tensile strength around 12 MPa, the amorphous material exhibits pronounced yielding and strain hardening, finally resulting in an elongation at break of over 200% and strength at break around 10 MPa. Thus, with no change in formulation composition and only a simple temperature adjustment, fundamentally different materials can be created.

## 3D printing within and above the liquid crystalline temperature range

Next, we translated the manufacturing of different parts with distinct material properties from one formulation from bulk curing conditions to 3D printing, utilizing different printing temperatures. Optimization of print irradiation parameters was performed at different temperatures ranging from 80 to 110 °C using an iterative process designed to emulate the irradiation settings for optimization of crystalline and amorphous properties previously described for bulk specimens. To verify these findings, a Jacob's working curve was recorded that confirmed sufficient curing depths (Supplementary Note 6). With the relatively high irradiation intensities used, we found near-complete curing directly after printing, and a post-curing step was not necessary (Supplementary Note 7). This was further confirmed through measurements of the polymers' gel fractions, which were above 97% for all tested print conditions (Supplementary Note 8). No significant differences in polymerization shrinkage were found, as all printing conditions resulted in a shrinkage between 2.5 and 3% (Supplementary Note 9).

To monitor the crystallinity response to a variety of printing conditions, the layers were irradiated at two different irradiation settings along with a temperature gradient from 78 °C to 110 °C, increasing the printing temperature by 2 °C every four layers (Fig. 3A). The same total light dose was emitted for each layer, with one half irradiated for 4.7 s at 80 mW cm$^{-2}$ and the other half irradiated for 24 s at 15.75 mW cm$^{-2}$. Analysis of the cross-section of the printed part using polarized optical microscopy revealed relatively clear-cut switching between bright crystalline and dark amorphous sections at 85 °C (80 mW cm$^{-2}$) and 87 °C (15.75 mW cm$^{-2}$). In addition, one layer printed at 82 °C did not crystallize when using a higher irradiation intensity. This is caused by too much heat development of the thiol-ene polymerization at this light intensity, causing the formulation to locally heat above the liquid crystalline temperature range. Therefore, the lower-intensity irradiation setting was used to reliably print crystalline specimens. The bright phenomena above 100 °C are directional traces of the microtome cut and not crystalline sections, which appear in different areas of the image, depending on the angle of the slice compared to the polarizing filters (Supplementary Note 10). The experiment also serves as a proof of principle for crystallinity manipulation using greyscale printing only, without changing the printing temperature. In a further experiment, using a constant temperature of 85 °C, the degree of crystallinity could be controlled simply by changing the irradiation intensity and thereby the exothermal response of the formulation (Supplementary Fig. 19).

Using a selection of settings from the temperature gradient print, a differentiation between opaque, crystalline and transparent, amorphous printed specimens made it evident that the multi-material effect can be achieved just as pronounced during printing as in bulk (Fig. 3B). The printing temperature of 110 °C, which was initially chosen for the isotropic liquid formulation state, resulted in relatively weak, soft specimens, some of which exhibited cracks upon removal from the platform. To explore this behavior further, an additional printing temperature of 100 °C was chosen, which also resulted in transparent, yet much more resilient specimens. Thus, three printing temperatures with corresponding irradiation settings were evaluated: 80 °C for printing from the liquid crystalline regime, 110 °C for printing from the isotropic melt, and an intermediate temperature of 100 °C, still exhibiting some crystallinity but resulting in a soft, transparent specimen. The chosen conditions are marked by colored rectangles in Fig. 3A.

DSC of the samples printed at 80 and 100 °C revealed distinct melting peaks (Fig. 3C), which result in high melt enthalpies ΔH (23.2 and 14.9 J g$^{-1}$, respectively, Supplementary Table 3). A melting point depression of 35 °C occurred for specimens printed at 100 °C compared to those printed at 80 °C. Interestingly, despite this distinct melting point, the specimens printed at 100 °C are highly transparent, as analysis of the optical properties revealed a large difference in optical attenuation ($\alpha = 2680$ m$^{-1}$ for 100 °C vs. $\alpha = 10640$ m$^{-1}$ for 80 °C, Supplementary Fig. 20). This behavior suggests that curing close to the LC-isotropic transition temperature still leads to some alignment of the liquid crystalline building blocks. The sample printed from the isotropic state at 110 °C exhibited some minor endothermic heat flow over a broad range from 75 to 100 °C, indicating only traces of crystallinity also observed in the bulk sample polymerized at 110 °C.

As can be seen in DMA measurements (Fig. 3D and Supplementary Fig. 11), the sample printed at 80 °C exhibits increased stiffness due to crystallinity, maintaining a high storage modulus between room temperature and 100 °C. Tensile tests also confirm a much higher stiffness of the semicrystalline specimens 3D printed at 80 °C compared to samples printed at 100 and 110 °C (Fig. 3E). Compared to the bulk cured crystalline sample, only slightly lower tensile strength was achieved, while the elongation at break was doubled. While the amorphous specimens printed at 110 °C did not achieve satisfactory results compared to the amorphous bulk polymer, the adjustment to 100 °C printing temperature produced a polymer network that is still soft, yet tough, probably due to minor alignment of the liquid crystalline building blocks, which is still possible at this temperature.

Printing of a five-layer composite structure, where the layers are printed in turns at 80 and 100 °C, gave the same melting temperature region in DSC measurements compared to the samples entirely printed at 80 °C, albeit with a smaller signal. This indicates that the layered structure did not impede the formation of crystalline and amorphous material properties in adjacent layers. The tensile properties of the composite structures were found to be between the amorphous and crystalline samples, which verifies good layer adhesion between crystalline and amorphous layers and demonstrates the capability for property tuning through multi-material printing.

We tested the formulation's capabilities to be 3D printed into a complex hollow pyramid shape. As this experiment required an extended amount of printing time due to the specimen's number of layers (140 layers compared to e.g., 40 for a tensile test specimen), we enhanced the formulation's stability by adding a larger amount of the radical stabilizer pyrogallol (0.5 wt% instead of 0.2 wt%). The resulting loss in reactivity was compensated by increasing the amount of the photoinitiator TPO slightly (1.5 mol% instead of 1 mol%). This adjustment, along with an intermediate printing temperature of 90 °C, caused slight crystallinity to occur in the pyramid, making it semi-opaque. To minimize printing time, we employed the short, high-intensity irradiation program previously used for amorphous parts (4.7 s, 80 mW cm$^{-2}$). Layer formation throughout the printing process

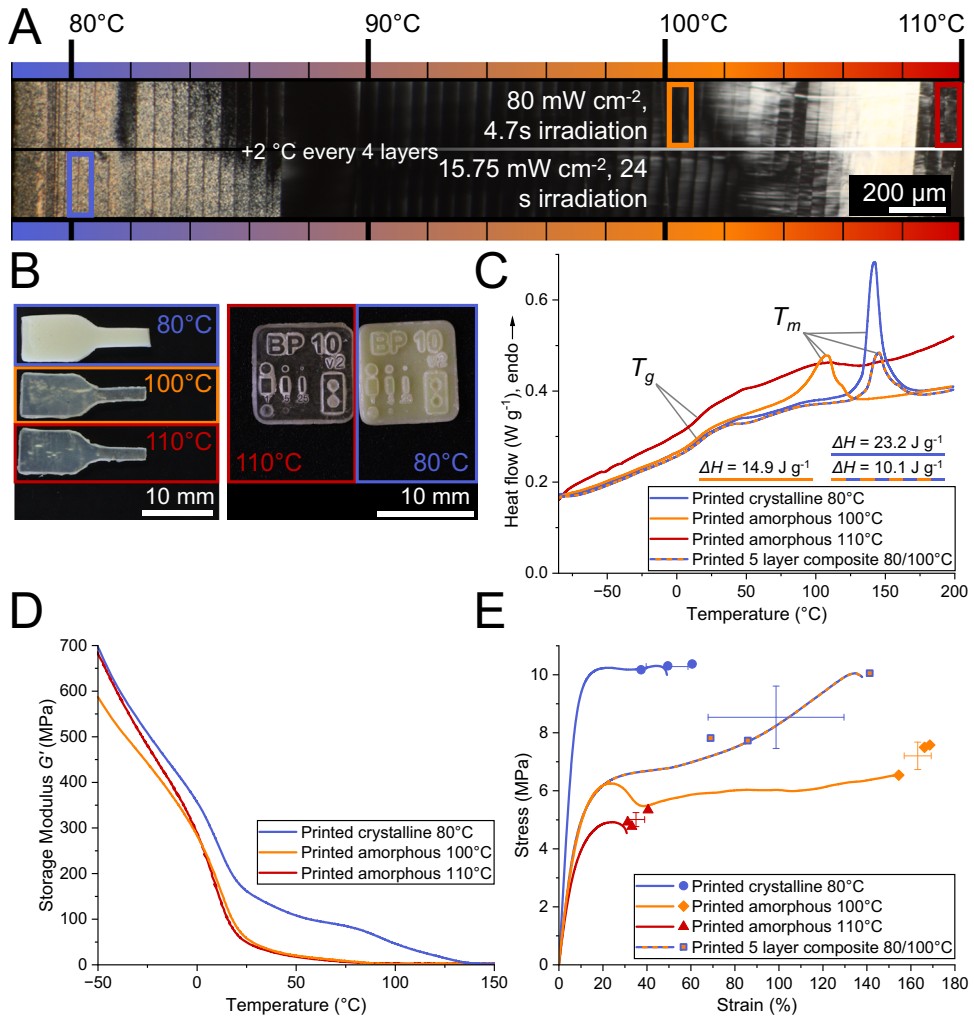

**Fig. 3 | Variation of material properties in printed parts depending on curing conditions. A** Polarized optical microscope image of a microtome cut of a polymer specimen cured under a temperature gradient at two different irradiation conditions. Throughout the printing process, a temperature gradient from 78 to 110 °C was applied by increasing the temperature by 2 °C every four layers. The specimen was cured at two different light intensities and irradiation times. Each layer was irradiated for 4.7 s at 80 mW cm$^{-2}$ on one half of the print, and for 24 s at 15.75 mW cm$^{-2}$ on the other half (same total light dose). The bright phenomenon between 105 and 108 °C is caused by the tracing of the microtome knife. Which areas turn bright depends on the angle of the polarizer (for comparison, see Supplementary Fig. 18). For (**B**–**E**), the curing temperature was varied between 80 °C (blue line), 100 °C (orange line) and 110 °C (red line). The samples cured at 100 and 110 °C were irradiated for 4.7 s at 80 mW cm$^{-2}$, while the samples cured at 80 °C were irradiated for 24 s at 15.75 mW cm$^{-2}$ to facilitate crystallization. **B** Images of bulk cured specimens highlighting differences in sample transparency depending on curing conditions. Images were recorded at room temperature after post-processing. **C** Differential scanning calorimetry (DSC) with melt enthalpies ΔH. **D** Thermomechanical properties measured using dynamic mechanical analysis (DMA). **E** Selected tensile test curves with stress and strain at break from all tested specimens. The error bars display the mean ± SD, $n = 3$. Tensile test results include a five-layer composite sample, in which two amorphous layers are encased between crystalline layers (blue-orange dashed line). All (thermo)mechanical test specimens were printed in XY orientation (flat on the build plate). 3D models of all printed parts are displayed in Supplementary Fig. 3.

was successful and resulted in an even and distortion-free formation of the pyramid shape (Fig. 4A). While minor overpolymerization was observed in certain areas of the inside of the beams (Fig. 4B), the scanning electron microscope image demonstrates pixel-perfect curing and even layer dimensions and spacing (Fig. 4C). Therefore, the appearance of overcuring was more attributed to insufficient post-processing for removal of oligomeric formulation, which is solid at room temperature.

**Multi-property printing**
After the successes in modulating the transparency and mechanical properties during the printing process, we expanded the technique to produce structures with two distinct material properties in each layer. The process to print a layer begins by preheating the build plate and the vat to 100 °C for curing the amorphous sections of the material. Subsequently, the build plate is lowered into the vat, and the amorphous sections are cured selectively. This is followed by a cooling step to 80 °C, after which the crystalline sections are cured (Supplementary Note 12).

To demonstrate the precise spatial control over the material properties, we printed a QR code from crystalline and amorphous sections (Fig. 5A). For imaging, transmitted light microscopy was utilized, in which amorphous sections appear white due to transmitted light and crystalline sections appear dark because less light penetrates the sample. Visual inspection of the 3D printed part under transmitted light microscopy shows that the material properties can be defined for each printed voxel (volumetric pixel). Each voxel has a size of 50 x 50 x 50 μm³ according to the capacities of the DLP light engine with a pixel pitch of 50 μm and a layer height of 50 μm. The magnified detail clearly showcases that the definition of the material property per voxel is possible. In addition, we illustrate a 3D printed "body with skeleton" model with a fine-grained control over the material

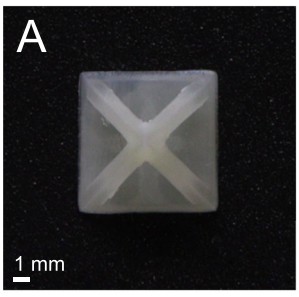
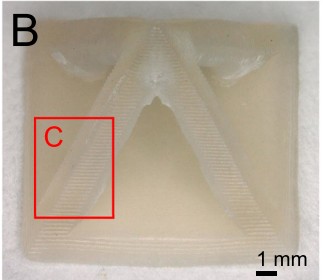
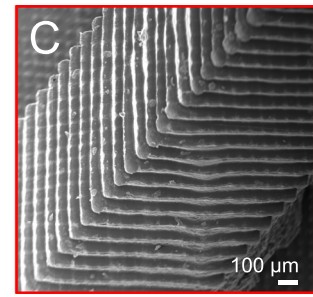

**Fig. 4 | Printed hollow pyramid specimen showcasing high resolution and even layer formation of the printed parts.** A formulation containing an increased amount of the stabilizer pyrogallol (0.5 wt% instead of 0.2 wt%) and a slightly increased amount of the photoinitiator TPO (1.5 mol% instead of 1 mol%) was printed into a challenging "additive manufacturing only" hollow pyramid shape with a 7 × 7 mm baseplate. **A** Camera image from above. **B** Digital microscope image. **C** Scanning electron microscope image of the area highlighted with a red rectangle in (**B**). 3D models of all printed parts are displayed in Supplementary Fig. 3.

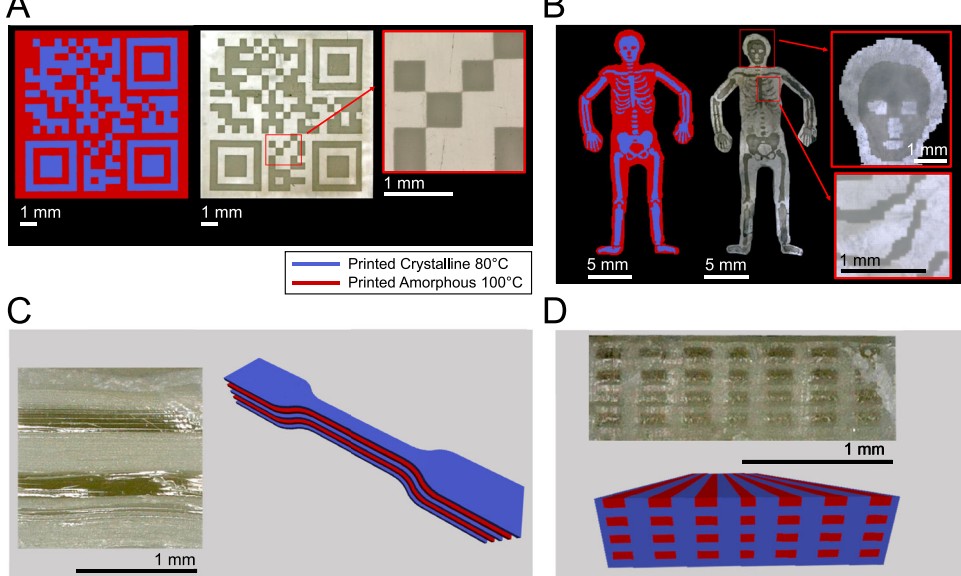

**Fig. 5 | Samples of 3D printed parts with different material properties depending on the curing conditions.** Parts of the models printed at 80 °C are shown in blue, parts printed at 100 °C in red. **A** Digital design of a QR code, the 3D printed object visualized under transmitted light microscopy, and a magnified detail of the printed code. **B** Digital design of a "body + skeleton" model with two properties, its overall appearance under transmitted light microscopy, and magnified details of the head and rib regions. **C** Cross-section of a 3D printed five-layer tensile test specimen with three crystalline and two amorphous layers and the 3D design corresponding to the printed specimen. **D** 3D printed cross-section of a fiber composite structure with amorphous toughening elements in the crystalline matrix and visualization of the corresponding voxel-based design. 3D models of all multi-material printed parts are displayed in Supplementary Fig. 4.

properties (Fig. 5B). For this part, an encapsulation of the intricate details of the skeletal model was achieved by embedding the layers strategically with layers of amorphous material. The two magnified detail images show that the transparency of the amorphous layer exhibits minimal obscuration of the fine details inside the model. In our experiments, we found the dimensional accuracy of crystalline-amorphous differentiation to be very good with a maximum deviation of 22 μm from the original design (Supplementary Note 13). Using this technique, composite structures can be manufactured. Figure 6C, D showcase the capability of this printing technique to produce composite structures of any desired geometry and soft/hard material composition, of which the five-layer composite was printed as a full tensile test specimen and tensile tested (Fig. 3E).

## Functional applications

The unique combination of tunable mechanical properties and the simultaneous presence of crosslinking and crystallinity opens many smart material applications for this monomer combination. Shape memory behavior was confirmed by cyclic testing using a dynamic mechanical analysis instrument (Fig. 6A and Supplementary Note 14). Herein, we performed 8 cycles of heating, elongation to 50% strain, cooling, shape fixity evaluation by releasing tension and shape recovery by heating under no tension. The shape fixity ratio demonstrates the material's ability to be fashioned into an arbitrary shape without reverting back to its original shape in the absence of the shape recovery stimulus (here: temperature increase). The specimens expanded slightly upon cooling, which points to a negative thermal expansion coefficient due to crystallization from the elongated state, as reported previously for liquid crystalline polymer networks[35]. This expansion compensates for the contraction that usually occurs upon cooling, finally resulting in a "slightly above perfect" average shape fixity ratio of 100.2%. The shape recovery ratio is the percentage of elongation recovered by heating the specimen to its crystallite melting point[36]. Due to the so-called training phenomenon[36], a reduced shape recovery rate of 80% was achieved in the first cycle. In subsequent cycles, more than 96% shape recovery ratio was achieved, and from the fifth cycle onwards, the shape recovery ratio reached a near-perfect 99%. Throughout all measured cycles, a satisfactory average recovery

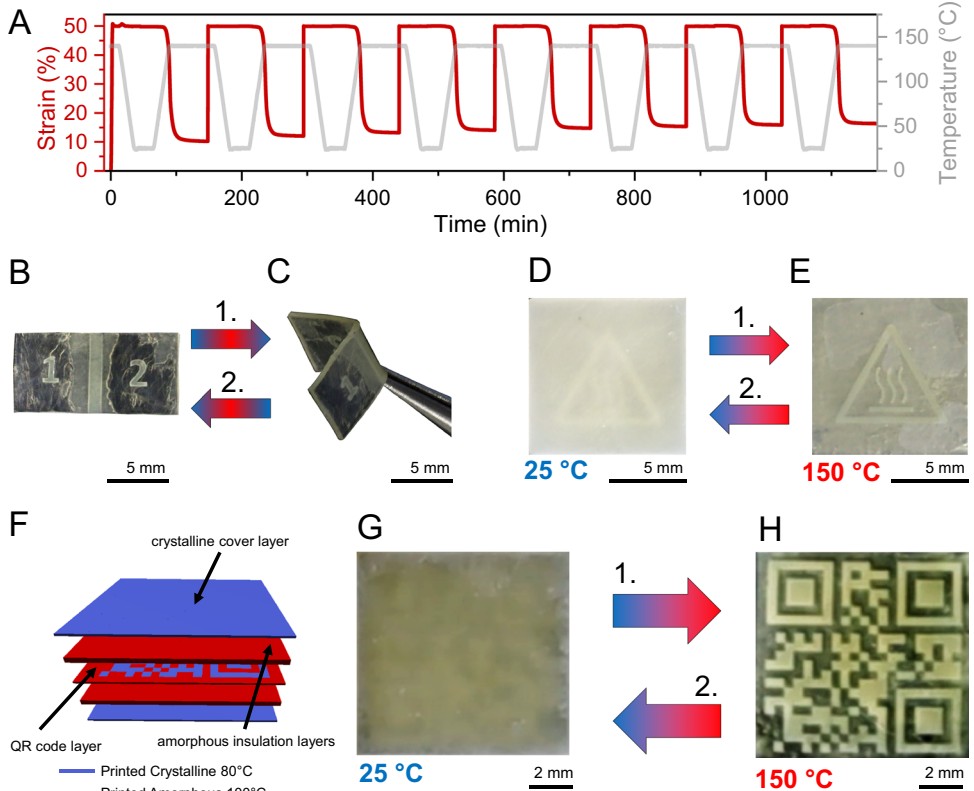

**Fig. 6 | Potential of the material to be used for stimuli-responsive functional applications. A** Cyclic shape memory testing of a bulk cured specimen exhibiting near-perfect shape fixity and shape recovery. Strain is depicted as a red line, temperature as a gray line. **B, C** A 3D printed shape memory part that exhibits shape programmability in a defined crystalline area. **D, E** A 3D printed heat warning sign is revealed at high temperature due to the melting of crystallites in a thin 3D printed layer and obscured again when cooled. **F–H** 3D printed part that illustrates the process of storing data with the help of controlled formation of crystalline and amorphous material. **F** The cross-sectional view of the voxel-based model of the printed structure shows a QR-code layer, crystalline cover layers, and amorphous insulation layers in between. Crystalline areas printed at 80 °C are depicted in blue, amorphous areas printed at 100 °C in red. **G** At room temperature (25 °C), no QR code is visible due to the crystalline cover layer. **H** When the top layer is evenly heated to 150 °C, the QR code is revealed. Removing the heat source causes the part to change back to its uniform opaque appearance. 3D models of all multi-material parts are displayed in Supplementary Fig. 4.

rate of 96% was achieved. In another measurement, in which a slightly modified procedure was used (which used the same procedure for the first cycle), the training phenomenon did not occur during the first measurement cycle, indicating that with further optimization of the polymerization procedure, near-perfect shape recovery could be achieved. This measurement also proved a single specimen's ability to be subjected to thirty measurement cycles without fracture (Supplementary Table 5 and Supplementary Figs. 24–26).

With the unique capability of crystallinity control during 3D printing, specimens with crystalline, shape-imprintable and recoverable sections can be created, while other sections are deliberately left amorphous. Hereby, smart micro-devices can be fabricated without the need to join different materials through mechanical or chemical techniques. A proof of concept is presented in Fig. 6B, where a printed part has a crystalline joint section and the rest of the part is amorphous. By heating the object, the crystalline joint was bent to the desired angle, which is fixed upon cooling to room temperature. Upon heating the sample again, the initial shape was recovered.

Beyond the use of the monomer combination's tunable thermomechanical properties, the adjustable optical attenuation in the visible range of light presents options for parameter-dependent encoding of information. For example, this could be applied in the form of a multi-material printed heat warning sign that only becomes visible above a certain temperature (Fig. 6C). Herein, a crystalline top layer renders the sign indiscernible at room temperature. Another advanced application would be encrypted chemical information storage. For this

application, we printed a five-layered part that consists of a QR code obscured by crystalline layers and insulated by amorphous layers (Fig. 6D and Supplementary Movie 1). At room temperature, the code is invisible and can only be revealed by placing a glass lid on top of the printed part and heating the lid. This enables a targeted heat transfer to the top layer of the part, causing it to turn isotropic and rendering the code readable. When heat is applied without the correct set of instructions (too harshly or not specifically at the top layer, i.e., in an oven or with a heat gun without a glass lid), parts of the QR code also disappear, rendering it impossible to scan and thereby increasing data storage security.

In conclusion, the ability to program mechanical and optical properties freely in three-dimensional space holds great potential for highly advanced applications. We present a noncomplex photopolymerizable system, which, through a simple printing parameter adjustment, can be cured to yield radically different materials by selectively translating a liquid crystalline phase into a semicrystalline polymer network. We were able to fully reproduce and enhance the results of the bulk material study in a 3D printer by depositing voxel-accurate crystalline and amorphous polymer sections in 3D. Thus, we established multi-material printing from a single formulation, which we used to create composite layers and fiber structures, shape memory materials, and parts capable of storing and selectively displaying encrypted information. First smart applications of this monomer system have been explored in this work, and we believe that the principle may be translatable to other monomer systems to further expand the scope of applications.

## Methods

### Materials

4,4′-Dihydroxybiphenyl (99%) was purchased from Fisher scientific. Anhydrous tetrahydrofuran was taken from a PURESOLV™ plant from Innovative Technology®. 10-Undecenoyl chloride (99%) was purchased from BLDpharm. Thioacetic acid (96%) was purchased from TCI. 2,2-Dimethoxy-2 phenylacetophenone (IRGACURE 651) and tetrahydrofuran (98%) were purchased from Fisher Chemical. Aqueous HCl (conc.) was purchased from VWR chemicals. NaCl was purchased from Donauchem. Diethyl ether (99.8%), sodium hydroxide (98%), sodium sulfate (anhydrous), triethylamine (99%), ethanol (96%), toluene (> 99.7%) 1,2,4,-trivinylcyclohexane (mixture of isomers, 98%) were purchased from Sigma-Aldrich.

### Instruments

To analyze the synthesized monomers, a Bruker Avance DRX-400 FT-NMR spectrometer at 400 MHz for $^1$H and 100 MHz for $^{13}$C was used. The spectra were referenced to the NMR solvent ($^1$H, chloroform-D$_1$: 7.26 ppm; $^{13}$C, chloroform-D$_1$: 77.16 ppm; 1H, dimethyl sulfoxide-d$_6$: 2.5 ppm). Spectra were evaluated using the software MestreNova.

To perform polarized optical microscopy, a Zeiss Axio.Scope A.1 microscope, combined with a Linkam LTS 350 heated stage, was utilized. Images were recorded using a Canon EOS 250D DSLR camera, which was attached to the microscope using a custom-made connector manufactured by RafCamera.

Melt rheology was performed using an Anton Paar MCR 300 rheometer.

Bulk polymers were cured using the heated polymerization chamber described in detail in the supplementary information of our previous work[24].

Differential scanning calorimetry (DSC) was measured using a TA Instruments DSC 2500 and recorded using the software TA Instruments TRIOS.

Dynamic mechanical analysis (DMA) was performed using an AntonPaar MCR 302 instrument equipped with an SRF12 rectangular fixture. Data were recorded using the software AntonPaar RheoPlus.

Tensile tests were performed using a Zwick Z050 and recorded using the software Zwick testXpert.

Real time photorheology was performed using an Anton Paar MCR 302 rheometer coupled to an Omnicure S2000-XLA mercury light source with a 320−500 nm band pass filter.

3D printing was performed by using a special form of digital light processing (DLP) 3D printing, namely hot lithography. With the prototype machine, BP10, it is possible to reach temperatures of up to 150 °C. A CAD drawing of this machine is provided in Supplementary Fig. 1. For image projection, we utilized an Invision Ikarus II light engine, equipped with a UV LED light source capable of projecting images with up to 80 mW cm$^{-2}$ with a pixel pitch of 50 by 50 μm. The emission wavelength is centered around 385 nm (Supplementary Fig. 2). For heating the vat, an IR heating system was installed below the vat. The build plate and frame of the vat are also both heated. A fan was used to cool the vat rapidly from below when printing multi-material parts. The build plate and attached part were not cooled with a fan because of the potential distortion that may have happened. For heating the aluminum build plate from 80 °C to 100 °C, a heating rate of ~ 8 °C min$^{-1}$ was achieved, and the cooling rate was around 7 °C min$^{-1}$. The IR heating system of the vat achieved a higher heating rate of ~23 °C min$^{-1}$, and with the enabled cooling fans, a cooling rate of approximately 20 °C min$^{-1}$ was reached. The cooling fans are enabled when the target temperature is below the measured temperature of the vat. Faster cooling of the build plate was not implemented because it would have required modification of the construction of the build plate locking system. The custom-written code for the multi-material printing process is available on TU Wien Research Data (see Code Availability Statement)[37].

The sample printed with a temperature gradient was sliced using a MICROM HM 360 rotational microtome equipped with a CN 30 cryo system.

Scanning electron microscopy was performed using a Zeiss Evo 10 microscope.

Digital light microscopy was performed using a Keyence VHX-6000 digital microscope.

ATR-IR spectroscopy was performed using a PerkinElmer Spectrum 65 FT-IR Spectrometer equipped with a Specac MKII Golden Gate Single Reflection ATR System. Results were recorded and evaluated using the PerkinElmer Spectrum software.

The transmittance and attenuation coefficient of the 3D printed polymer samples were measured using a Bruker Vertex 80 v Fourier-transform infrared (FTIR) interferometer. The samples were flat rectangular chips measuring 8 × 8 mm² with a thickness of 0.5 mm. The measurements were performed in transmission mode at room temperature. The samples were mounted on a copper plate with a 5 mm aperture. Reference measurements were performed using the blank copper mount with a 5 mm aperture. From 10000 nm to 1250 nm, the radiation source used was a silicon carbide globar with a potassium bromide beam splitter and a deuterated triglycine sulfate (DTGS) infrared detector. From 1250 nm to 400 nm, a tungsten bulb with a calcium fluoride beam splitter and a silicon diode detector were utilized. Measurements were taken with a 10 cm$^{-1}$ resolution. The data was recorded with OPUS software, and analysis was performed with Wavemetrics IgorPro software, including an interpolation of the 10 cm$^{-1}$ resolution data to 1 cm$^{-1}$ point spacing, calculating transmission and attenuation coefficient.

Shape memory tests were performed using a TA Instruments DMA 850 (for measurements according to literature procedures) or RSA-G2 dynamic mechanical analysis instrument (for adapted measurements) equipped with a tensile test sample fixture. The results were recorded using the software TA Instruments TRIOS.

Measurement graphs were created using OriginPro 2023b. Microsoft Excel was used for calculations. Averages of measurement values are given as the arithmetic mean. Standard deviations were calculated using Microsoft Excel's *stdev.p* function. The printed single-material models were modeled using Autodesk Fusion 360. Slicing of 3D models was performed using PrusaSlicer. The multi-material parts were drawn layer by layer using MS Paint and read out using the code available in the research data. Visualization of the voxel models of the multi-material parts was performed using a voxel-based software based on a master's thesis project[38].

### Monomer synthesis

4′-(9-Decenylcarbonyloxy)−4-biphenylyl 10-undecenoate (BPLC) was prepared according to Imae et al.[39].

4,4′-Dihydroxybiphenyl (1 eq, 0.0705 mol, 13.13 g) and triethylamine (2.2 eq, 0.1551 mol, 15.69 g) were dissolved in 300 mL THF and stirred under argon atmosphere in a 500 mL round bottom flask. The mixture was cooled to 0 °C and 10-undecenoyl chloride (2.1 eq, 0.1445 mol, 30.02 g) was added slowly via dropping funnel while keeping the mixture at 0 °C, after which the reaction mixture was allowed to warm up to room temperature. After stirring for another 48 h, the mixture was filtered through a thin layer of silica, which was washed with dichloromethane. A rotational evaporator was used to remove the solvent from the filtrate. The crude product was recrystallized from a small amount of ethanol and dried under vacuum, which afforded 34.05 g (92%) of the product as white flakes.

$^1$H NMR (400 MHz, CDCl$_3$) δ = 7.57 − 7.52 (m, 4H, Ar-$\underline{H}$), 7.18 − 7.08 (m, 4H, Ar-$\underline{H}$), 5.89 − 5.70 (m, 2H, =C$\underline{H_2}$), 5.03 − 4.91 (m, 4H, -$\underline{CH}$ = ), 2.58 (t, J = 7.5 Hz, 4H, -C$\underline{H_2}$-), 2.06 (tdd, J = 6.6, 5.3, 1.4 Hz, 4H, -C$\underline{H_2}$-), 1.83 − 1.71 (m, 4H, -C$\underline{H_2}$-), 1.37 (m, 16H, -C$\underline{H_2}$-).

$^{13}$C NMR-APT (101 MHz, CDCl$_3$) δ = 172.51 (-$\underline{C}$ = O), 150.37 ($\underline{C}_{Ar}$-C = O), 139.32 (-$\underline{C}$H = ), 138.21 ($\underline{C}_{Ar}$-H), 128.27 ($\underline{C}_{Ar}$), 122.05 ($\underline{C}_{Ar}$), 114.32

( = $\underline{C}H_2$), 34.58 (O-$\underline{C}H_2$-CH), 33.94 (-$\underline{C}H_2$-), 29.44 (-$\underline{C}H_2$-), 29.36 (-$\underline{C}H_2$-), 29.24 (-$\underline{C}H_2$-), 29.20 (-$\underline{C}H_2$-), 29.05 (-$\underline{C}H_2$-), 25.10 (-$\underline{C}H_2$-).

1,2,4-Cyclohexanetriethanethiol (CHTT) was synthesized in analogy to our previous work[36].

1,2,4-Trivinylcyclohexane (mixture of isomers, 40 g, 246.52 mmol, 1 eq) was prepared in a 250 mL round bottom flask under argon atmosphere and cooled to −10 °C in an ice−NaCl bath. While continuously stirring and keeping the mixture around −10 °C, thioacetic acid (60.05 g, 788.86 mmol, 3.2 eq) was added slowly via dropping funnel. After 30 min, three portions of 2,2-dimethoxy-2 phenylacetophenone (IRGACURE 651, total amount 1.6 g, 6.24 mmol) were added. After each portion, the reaction mixture was irradiated for 20 min using UV light (Omnicure S2000; 320–500 nm filter; 110 mW cm$^{-2}$ at the source) through a Lumatec Series 300 liquid light guide, which was placed onto a quartz glass window inserted into a flask neck. The mixture was diluted using tetrahydrofuran (600 mL) in a 2 L round bottom flask, after which the solution was cooled to 0 °C and 2 M NaOH (560 mL) was added. The mixture was allowed to warm up to room temperature after 30 min and stirring was continued for another 48 h before neutralization with approximately 300 mL 4 M HCl, during which the mixture was cooled. Thereafter, three 200 mL portions of diethyl ether were used to extract the mixture. 150 mL brine was used to wash the combined organic phases, after which they were dried using sodium sulfate and concentrated using a rotary evaporator. $^1$H-NMR of the crude product revealed that the acetate peak (2.22 to 2.37 ppm) had fully disappeared, and the crude product was obtained as an orange oil. After distillation at 145 °C at a pressure of 0.022 mbar, 47.1 g (77.5%) of the product 1,2,4-cyclohexanetriethanethiol (CHTT, mixture of isomers) was obtained as a clear liquid with a slight thiol odor.

$^1$H NMR (400 MHz, CDCl$_3$) δ = 2.69 − 2.36 (m, 6H), 1.90 − 0.57 (m, 18H).

## Formulation preparation

To prepare the photopolymerizable liquid crystalline monomer formulation, the inhibitor pyrogallol and the photoinitiator diphenyl (2,4,6-trimethylbenzoyl)-phosphine oxide (TPO) were weighed into a brown glass vial. Next, the trifunctional thiol CHTT was added, and the mixture was heated to approximately 50 °C and stirred well using a vortex mixer. Lastly, the liquid crystalline monomer BPLC was added. The component ratios for the two different formulation compositions used in this work are listed in Supplementary Table 1. The mixture was heated to approximately 120 °C (isotropic melt) and homogenized using a vortex mixer by stirring for ~2 min. The formulation's melting point and liquid crystalline behavior were analyzed using polarized optical microscopy.

## Formulation stability

The temperature stability of the formulation containing 0.2 wt% pyrogallol and 1 mol% TPO was tested for its temperature stability using melt rheology. Around 100 mg of the formulation was transferred onto the preheated measurement surface, and the measurement was started after a one-minute equilibration period. Between measurements, the formulation was stored in a closed glass vial under an air atmosphere at 110 °C. The measurements were performed at a temperature of 100 °C.

## Microscopy

Polarized optical microscopy was used to analyze the liquid crystalline phase present in the formulations (Supplementary Fig. 6). A heating and cooling rate of 5 °C min$^{-1}$ was used to analyze the transition temperatures. The temperature gradient 3D print was recorded using polarized optical microscopy at room temperature (Supplementary Fig. 18).

To analyze printed specimens, digital light microscopy using depth composition was used in reflected light and transmitted light modes.

For the pyramid specimen, scanning electron microscopy images were recorded with an electron high tension voltage of 10.00 kV at a working distance of 12.12 mm.

## Real-time near-infrared photorheology

Real-time NIR photorheology was performed at 80 °C and 90 °C. Approximately 170 μl of the molten formulation were poured onto the measurement platform, after which a 25 mm-diameter parallel plate rheology stamp was lowered to leave a gap of 0.2 mm. After a three-minute equilibration period, the measurement was started. Initially, rheology was recorded for 65 s, after which the irradiation process was initiated through the measurement platform. The formulation was irradiated for 300 s with a light intensity of 30 mW cm$^{-2}$ from a mercury lamp with a 320–500 nm band pass filter.

## Bulk curing procedure

Bulk polymer specimens for tensile tests and dynamic mechanical analysis were cured in a custom-built heated mold, which enables accurate temperature control[24]. The formulation was poured into ~2 mm thick molds cast from silicone. The mold was preheated to 80 °C for crystalline polymer synthesis and to 100 °C for amorphous polymer synthesis. For the "mild irradiation" crystalline specimens, an initial irradiation step was conducted by irradiating the formulation for 300 s at 1.5 mW cm$^{-2}$ using a 405 nm LED light source. Afterwards, a second irradiation was employed at 290 mW cm$^{-2}$ for 180 s using a 365 nm LED light source. For the "strong irradiation" and amorphous specimens, only the second step was used. The irradiation steps were repeated after the specimens were carefully turned upside down and placed back into the mold. An overview of the curing conditions is given in Supplementary Table 2.

## Differential scanning calorimetry

To perform differential scanning calorimetry (DSC), 3-5 mg of the polymer networks were weighed into aluminum DSC crucibles and first cooled to −90 °C, after which they were heated to 200 °C. Thereafter, the cooling and heating cycle was repeated. Heating and cooling rates of 10 K min$^{-1}$ were used. For the measurements displayed in this article, the first heating cycle was used, since the second cycle displayed minor shifts in the crystallite melting points after being heated to 200 °C. Therefore, we considered the first cycle to be more representative of the polymer network present at room temperature after curing.

## Dynamic mechanical analysis

For dynamic mechanical analysis (DMA), cuboid samples according to DIN EN ISO 6721 were cured in bulk and using 3D printing as described previously. The bulk specimens were lightly sanded to correct geometric irregularities. The samples were clamped into rectangular fixtures and subjected to torsional shear strain of 0.1% with a frequency of 1 Hz through a temperature range of −50−200 °C with a heating rate of 2 K min$^{-1}$.

## Tensile testing

To measure tensile tests, dogbone shape specimens according to ISO 527 test specimen 5b were cured in bulk using 3D printing. For each sample, three specimens were tested, and averages and standard deviations were calculated therefrom. The bulk specimens were sanded to correct slight geometric irregularities. Tensile tests were performed with a traverse speed of 5 mm min$^{-1}$. 3D printed specimens were strained parallel to their print layer orientation.

## 3D printing

3D printing was performed using lithography-based 3D printing via digital light processing (DLP) light engine (up to 80 mW cm⁻², 385 nm). The used prototype printer employs independent control of the vat, building platform and vat frame. Depending on the desired material properties, the temperatures of the vat and building platform were adjusted. For experiments in this work, temperatures between 78 and 110 °C were used. The vat frame was kept at a constant temperature of 80 °C. For each print layer, the building platform was lowered to the vat, keeping a 50 μm-high section coated with the liquid formulation. To achieve parts with multiple properties within a single layer, we first cured the crystalline sections at 80 °C at a lower irradiation intensity applied over a longer time (24 s at 15.75 mW cm⁻²). Afterwards, the temperature was changed, and the amorphous sections were cured, for which a higher irradiation intensity was used (4.7 s at around 80 mW cm⁻²). After each curing step, the part was peeled off the vat by lifting the build plate, which allows for more of the formulation to flow under the previously printed material. The process was repeated until the part was finished. Each layer printed this way takes a total time of the pure irradiation time, with an added three seconds for the peeling process and two seconds for the lowering of the platform. Therefore, for an amorphous layer, the total printing time was 4.7 + 3 + 2 = 9.7 s. For a crystalline layer, the printing time was 29 s.

With a voxel-based modeling software, we were able to create multi-material objects that were then saved as a set of images, wherein each pixel corresponds to one voxel. To print different properties, the printer uses one of these images for each layer. Depending on the color of each pixel, the software is programmed to detect which material property should be printed and choose the right printing condition.

To determine the right exposure settings, we used a calibration object that indicates if over- or underpolymerization is present. Because this test specimen (Resolution Test Chip, Main Part Fig. 3B) can be printed quickly, several iterations were performed until a satisfactory result was achieved. To assess the result of the printing test, we used the hourglass shape on the test chip, which has tips that should barely touch when neither over- nor underpolymerization is present. The final calibration settings were 15.75 mW cm⁻² for 24 s at 80 °C and 80 mW cm⁻² for 4.7 s at 100 °C and 110 °C. To confirm our findings, we conducted curing depth measurements on the BP10, which were performed by heating a small amount of the formulation to the desired temperature on the printing platform and then exposing it to UV light. Subsequently, the uncured material was removed, and the cured layer was taken out of the vat. The results of the curing depth measurement (Supplementary Fig. 12) displayed as a Jacob's working curve indicate that the described iterative search had indeed yielded a reliable result.

All mechanical and thermomechanical (tensile tests and DMA) test specimens were printed laying in the XY direction (flat on the build plate). An upright printing orientation was not feasible due to formulation stability (40 layers for a flat DMA specimen vs. 800 layers for an upright specimen). An upright printing direction would likely have resulted in a 15–35% decrease in tensile strength across all samples[40]. However, an effect on the unique properties presented in this work is unlikely.

For multi-material parts, we modified the previously described lithography-based printing process to be able to print crystalline and amorphous properties within a single layer: Supplementary Fig. 18 provides a graphical overview of the measured temperatures during the fabrication of a multi-material layer.

At the beginning of each layer, the build plate is lowered into the vat, leaving a 50 μm gap (2 s). The target temperature for the build plate and vat was set to 100 °C for printing of amorphous specimens. Reaching the target temperature takes approximately 148 s, and an additional waiting time of 2 to 10 s was implemented to reach thermal equilibrium. We tested waiting times between 2 and 10 s and selected a

duration of 5 s, as no noticeable effect of this parameter on the process could be observed. The first exposure step is carried out at an intensity of 80 mW cm⁻² for 4.7 s. To cure the material from the liquid crystalline range to obtain semi-crystalline polymer networks, the build plate and vat had to be cooled again to 80 °C (155 s), followed by another 5 s waiting period. The second exposure was carried out with 15.75 mW cm⁻² for 24 s. Finally, the freshly printed layer is peeled from the vat, taking an additional 3 s. Therefore, the total time required to print one multi-material layer sums up to 343.7 s.

## Post-processing

Post-processing of parts was performed by submerging the part in toluene and sonication for up to five minutes at around 35 °C. The post-processing procedure was optimized to avoid swelling, which usually caused cracks to form in the printed parts. For the optimization test, multiple specimens of the resolution test chips displayed in Fig. 3B were utilized. After the first two minutes of sonication, the parts were frequently removed from the solution, and the procedure was stopped before swelling of the main printed part, which was predictable as the oligomerized residues started swelling significantly before any swelling of the main part occurred. The crystalline parts were sonicated for 4-5 min, while the amorphous parts were sonicated for 2-3 min due to their lower solvent resistance and typically slightly less amounts of unpolymerized/oligomerized residue compared to the crystalline parts. The multi-material parts were post-processed the same way as amorphous parts. No change in transparency or mechanical behavior was observed as a result of the post-processing procedure. No further post-curing using UV irradiation was deemed necessary for mechanical tests, as high conversions and stable transparencies were achieved without post-curing, which is demonstrated in Supplementary Figs. 13–15 and 19.

Double bond conversions of printed specimens were investigated via ATR-IR spectroscopy and calculated via [1].

$$DBC = \left(1 - \frac{\left(\frac{DBI_{polymer}}{COI_{polymer}}\right)}{\left(\frac{DBI_{formulation}}{COI_{formulation}}\right)}\right) * 100 \qquad (1)$$

$DBC$ ...... double bond conversion (%)

$DBI_{polymer}$ ...... double bond integral of the measured polymer sample

$COI_{polymer}$ ...... carbonyl bond integral of the measured polymer sample

$DBI_{formulation}$ ...... double bond integral of the uncured formulation

$COI_{formulation}$ ...... carbonyl bond integral of the uncured formulation

## Gel fraction determination

To measure the gel fraction of the 3D printed polymers, they were weighed and subsequently submerged in toluene at room temperature for 24 h. For the experiment, non-post-cured samples were used to test for a possible incomplete network formation. After drying at 120 °C until a constant weight was reached, the dried sample mass was measured and the gel fraction was determined by dividing the dried mass by the original mass.

## Polymer density analysis

The densities of the formulation and of polymer samples 3D printed under different conditions were measured via the Archimedes method. The formulation could be measured using the Archimedes method, as it crystallizes into a waxy solid after melting. Densities were

calculated using equation [2].

$$\rho_s = \frac{m_1}{(m_1 - m_2) * \alpha} * \rho_{H_2O} + \rho_{air} \qquad (2)$$

$\rho_s$ ...... sample density (g cm$^{-3}$)
$m_1$ ...... sample mass
$m_2$ ...... sample mass submerged in water
$\alpha$ ...... weight correction factor = 0.99983
$\rho_{H_2O}$ ...... density of water at measurement temperature (25 °C, 0.997044 g cm$^{-3}$)
$\rho_{air}$ ...... density of air (0.0012 g cm$^{-3}$)

**Cryo-microtome sample preparation**
Cryo-microtome cuts of the temperature gradient 3D print were performed using a glass knife at a thickness of 10 μm at −80 °C.

**Shape memory testing**
Two different methods of shape memory evaluation were performed in this work. An eight-cycle test was performed using the literature-known conventional method[40], while three ten-cycle tests were performed on a different specimen, where a modified measurement procedure was used.

Cyclic shape memory tests were performed on dogbone shape specimens in analogy to those used in tensile tests. The specimens were fastened into a dynamic mechanical analyzer equipped with a rectangular fixture. First, the samples were heated to 140 °C using a heating rate of 20 K min$^{-1}$. After an equilibration period of 15 min, they were strained until the parallel section of the specimen had reached 50% elongation. Next, the samples were cooled to room temperature using air cooling with a cooling rate of 5 K min$^{-1}$. At room temperature, the tension applied by the instrument was released to measure the resulting contractive or expansive response of the samples, from which the shape fixity ratio was calculated. To test the shape recovery, the samples were again heated to 140 °C using a heating rate of 10 K min$^{-1}$ under no external stress, where they were held for 60 min. To calculate the shape recovery ratio, the contraction of the samples during this heating process was divided by their sample strain after being allowed to contract upon releasing the instrument's tension. After contracting for 60 min, the process was repeated by straining the sample again. The specimens were strained parallel to their print orientation. Here, using the conventional measurement technique, the sample was strained to the same absolute elongation reached previously. When measuring using the modified measurement method, the sample was elongated to 50% strain starting from the elongation reached after shape recovery, causing the sample to elongate throughout multiple measurement cycles. The shape fixity and shape recovery ratios are determined via equations [3] and [4][36]. The calculations are listed in Supplementary Tables 4 (conventional method) and 5 (modified method), and the measurement graphs are displayed in Supplementary Figs. 23 (conventional method) and 24–26 (modified method).

The shape recovery ratios were calculated using the elongation values before the relaxation of the sample in the elongated state, since in our case, this relaxation caused the samples to elongate and not contract. This leads to most shape fixity ratios resulting in values above 100%. By using the standard formula to calculate the shape recovery ratio via equation [5], the results would be falsely skewed toward higher recoveries. Where the shape fixity ratio was below 100%, [5] was used to calculate the shape recovery.

$$R_{sf} = \frac{L_r - L_0}{L_s - L_0} * 100 \qquad (3)$$

$R_{sf}$ ...... shape fixity ratio (%)

$L_r$ ...... parallel sample section length of the elongated and cooled sample after release of force *(mm)*
$L_O$ ...... parallel sample section length before elongation (mm)
$L_s$ ...... parallel sample section length after elongation while force is still applied (mm)

$$R_{sr} = \frac{L_s - L_1}{L_s - L_0} * 100 \qquad (4)$$

$R_{sr}$ ...... shape recovery ratio calculated in this work (%)
$L_1$ ...... parallel sample section length after recovery (mm)
$L_s$ ...... parallel sample section length after elongation while force is still applied (mm)
$L_O$ ...... parallel sample section length before elongation (mm)

$$R_{sr'} = \frac{L_r - L_1}{L_s - L_0} * 100 \qquad (5)$$

$R_{sr}{'}$ ...... shape recovery ratio calculated (%)
$L_r$ ...... parallel sample section length of the elongated and cooled sample after release of force (mm)
$L_1$ ...... parallel sample section length after recovery (mm)
$L_s$ ...... parallel sample section length after elongation while force is still applied (mm)
$L_O$ ...... parallel sample section length before elongation (mm)

**Characterization of functional multi-material parts**
The multi-material shape memory part (Fig. 6B, C) was shape imprinted by bending it at the crystalline middle section and briefly heating it to 150 °C using a heat gun and cooling it back to room temperature, upon which the new shape was fixed. By heating the part to 150 °C again, the original flat shape was recovered.

The temperature warning sign (Fig. 6D, E) was set up by placing a microscope glass slide under and over the printed warning sign. The printed part was placed on top, and another glass slide was used to cover the printed part. The specimen was heated using a heat gun.

The concealed QR code (Fig. 6F–H) was revealed by placing a glass slide on top of the specimen and heating it with a heat gun evenly from above. Below the specimen, a metal plate was placed to ensure a heat sink effect.

## Data availability
All data generated in this study have been deposited in the TU Wien Research Data database under accession code https://doi.org/10.48436/drx13-s7537[37]. In addition, all data are also available from the corresponding author upon request.

## Code availability
The code for all print jobs performed in the related publication and a corresponding readme-file are available at TU Wien Research Data at https://doi.org/10.48436/drx13-s7537[37]. In addition, all data are also available from the corresponding author upon request.

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

## Acknowledgements

This research was funded in whole or in part by the Austrian Science Fund (FWF) [10.55776/RIC9773224]. For open access purposes, the author has applied a CC BY public copyright license to any author accepted manuscript version arising from this submission.

## Author contributions

M.G.: Methodology, Validation, Formal Analysis, Investigation, Data Curation, Writing – Original Draft, Writing – Review & Editing, Visualization. D.L.: Methodology, Software, Validation, Formal Analysis, Investigation, Writing – Original Draft, Writing – Review & Editing, Visualization. T.K.: Methodology, Formal Analysis, Investigation, Writing – Review & Editing. E.C.: Methodology, Formal Analysis, Investigation, Writing – Review & Editing. X.L.: Investigation. A.P.: Formal Analysis, Writing – Review & Editing. J.S.: Formal Analysis, Resources, Writing – Review & Editing, Supervision, Funding Acquisition. R.L.: Formal Analysis, Resources, Writing – Review & Editing, Supervision, Funding Acquisition. K.E.: Conceptualization, Methodology, Validation, Formal Analysis, Investigation, Data Curation, Writing- Original Draft, Writing – Review & Editing, Visualization, Supervision, Project Administration.

## Competing interests

The authors declare no competing interests.
