## [Transparent Peer Review file · Nature Communications]

Semi-crystalline and amorphous materials via multi-temperature 3D printing from one formulation

Corresponding Author: Dr Katharina Ehrmann

Version 0:

Reviewer comments:

Reviewer #1

(Remarks to the Author)

In this study, multimaterial 3D printed parts using vat polymerization 3D printing is demonstrated to print both semicrystalline and amorphous polymers in spatially defined areas from a single liquid crystalline monomer resin. The amorphous vs. crystalline polymer regions are controlled via printing temperature and light intensity. There are a handful reports in literature related to vat polymerization of semicrystalline polymers, but this study is unique in terms of the use of liquid crystalline monomers for multimaterial printing and the ability to control between amorphous and semicrystalline polymers (with variations in opacity and mechanical properties) with temperature and light intensity in one print and within one layer, as opposed to switching materials during the print. However, there are flaws in the data analysis, some details are missing from the methods, and clarity of the manuscript needs to be significantly improved to better understand the results. The following points require major revision:

1. From the bulk and 3D printed results, the importance of both temperature and light intensity in controlling amorphous vs. semicrystalline domains is evident. However, in certain points throughout the text (e.g., the title that mentions only “greyscale printing” and page 6, paragraph 2 mentions “only a simple temperature adjustment”) only one or the other is mentioned. The importance of both should be more clearly stated. Also, the curing/printing conditions were sometimes hard to follow with either only temperature stated or vague light intensity conditions (e.g., “mild irradiation”). Perhaps a summary table would be useful.

2. The introduction is hard to follow and some arguments for why the proposed technique is a better multimaterial method are conflicting. For instance, it is said that exchanging the material feed is time intensive (page 2, paragraph 1), but changing the temperature and the long exposure times used in this study would also contribute to long print times. The literature examples on page 3, paragraph 1 were difficult to understand. The drawbacks of many other relevant multimaterial 3D printing examples were described in the introduction, but it wasn't always clearly stated how the method in this study is an improvement compared to these.

3. Page 4, paragraph 2 Property differentiation studies – Further support and experimental results should be provided as to why the previous formulation did not achieve property differentiation with printing parameters. Additional support for the choice of the new liquid crystalline ene-monomer (BPLC) should also be provided. Why would an even more highly ordered smectic X phase contribute to property differentiation with printing parameters? Discussion of previous studies (page 4, paragraph 3) is stated, but no reference or data is given.

4. In Figure 2D, please explain the standard deviation lines. How many repeat samples were tested?

5. While the bright phenomena above 100 degC was explained in Figure 3A as due to the way the sample was cut and the polarizing filters, it is misleading and makes it appear as if these areas are semicrystalline. Could this be avoided by printing a larger sample and having the cut parts outside of the optical microscope image? Or perform background image correction? Or include other images to prove the bright phenomena position changes with direction of the polarizing filters and isn't part of the sample? Also, the temperature scale is hard to follow (ticks of 8 degC). How many layers were printed at each temperature? Maybe a 3D model of the printed sample with different light intensities and temperatures described could be included to help explain the sample more clearly. There appears to be some bleeding of amorphous region into crystalline region between high and low intensity regions at ~86 degC. Is this real? If so, why?

6. Please provide a reference for the degree of crystallinity equation. The D_c is typically referenced to the 100% crystalline polymer – is it valid to use the BPLC monomer? It is stated that in addition to some bulk and 3D printed samples, a baseline correction could not be found for the BPLC pure monomer and the isotropic enthalpy could not be accurately calculated, yet

- it was still used as a reference point to calculate D_c for the bulk and 3D printed samples. Please explain how conclusions could be made with this method that could not be properly calculated. Page 6, paragraph 1 – please provide a reference showing that the broad flat peak at 100 degC for the amorphous sample signifies traces of crystallinity. How do these crystallinity values compare to other 3D printed semicrystalline or liquid crystalline polymers, especially in 3D printing?
7. The claimed "... pixel-to-pixel precision property switching within one printing layer ..." (Abstract, page 1; Introduction, page 3, paragraph 1; page 10, paragraph 3) is not clearly demonstrated, especially when referenced to the QR code in Figure 5A where the smallest feature is ~300 μm .
 8. Shape memory tests – although addressed with the adjusted shape recovery ratio equation, it is difficult to make the conclusion of outstanding average recovery and fixity when the samples elongate over time and do not contract as is typical. Is this common for liquid crystal polymers? More explanation and comparison to other relevant examples is needed to understand these results.
 9. The reference to the figures is out of order and/or not in logical locations in the text (e.g., Figure 1A, Figure 2A, Figure 1B, Figure 2B, Figure 3A, Figure 2C, etc.). Also, reference to some Supporting Information calculations, figures, and tables are missing and would be helpful to include in the text to provide further clarification on certain points (e.g., Section 6 Polymer crystallinity calculation, SI Figures 4 – 9, and SI Table 2)
 10. Some of the figure captions are not clearly described (e.g., Figure 2 and 3). Separate sentences should be written out for each part of the figure and include more details. Also, "thermomechanical" properties from tensile tests is incorrect if the samples were tested at room temperature (Figure 2 and 3 caption).
 11. There is confusion between the five-layer tensile and fiber structure in SI Figure 6 (the fiber tensile test 3D model looks like it is labelled as the five-layer tensile test in the SI). The total 3D model and cross-section models for certain prints should be grouped more clearly together (e.g., hidden QR code, fiber tensile test, five-layer tensile test, etc.) and both in the same document (main document or SI).
 12. It is confusing to have some Methods information in the main document and others in the SI or in both (e.g., 3D printing). Choose one or the other. An image of the custom 3D printer and the equipment used to heat vat would be useful. Why were different wavelengths used for the bulk curing and 3D printed samples? Page 14, 3D Printing – "3D printing was performed using stereolithography (SLA) with a digital light processing (DLP) light engine ..." SLA and DLP are two different printing techniques (rastering laser vs. 2D projected light exposure). Please clarify.
 13. Page 15, paragraph 1 (3D printing method) – Post-processing conditions were different for crystalline and amorphous parts. What post-processing was used for multi-material prints? The amorphous parts had less solvent resistance. Does this mean the part degraded more with the toluene wash for multimaterial parts? Could this have contributed to the change in mechanical properties? Was the difference in transparency between amorphous and crystalline prints similar before and after post-processing in toluene?
 14. What were the transmission and opacity for samples cured/printed at different light intensities?

Reviewer #2

(Remarks to the Author)

[Editor Note: See attached for Comments]

Reviewer #3

(Remarks to the Author)

The authors present a system to fabricate varied mechanical properties using a single vat of material by manipulating the vat temperature during polymerization and, in one experiment, greyscale (grayscale).

While the manuscript presents noteworthy results, there are some substantial revisions that would be required for acceptance.

- 1) The authors used bulk polymerization to determine the mechanical properties, which is known to produce significantly different results than specimen polymerized in a 3D printer. These mechanical properties would need to be repeated using 3D printed specimen (with varied print orientations given properties are known to be dependent on these conditions)
- 2) Given the title of the manuscript explicitly references the use of greyscale printing (in the pdf, which is different than the title listed here), more thorough investigation into the effects of printing these structures under varied intensities is required in addition to the thermal work. Given the simplicity of changing the greyscale in a given projected layer, this seems like it would have been the first, simplest consideration, which leads me to infer that the authors encountered issues with these efforts that are not adequately disclosed/discussed.
- 3) Given the breadth of novel chemistries being developed to produce a range of mechanical properties from a single 'pot' of resin in a given print, both a more thorough, recent literature review of these efforts is required. Also, further demonstration of the utility of this technique/approach is required.
- 4) It is concerning that the authors did not conduct an additional post curing steps after polymerization given the photoinitiator used is known to be a photoabsorber which would produce a gradient in conversion (thus properties) both within and, likely, between the printed layers. The only remedy for this is to introduce an additional curing step after the part has all excess monomer removed. More explanation as to why the post-curing step was not conducted is required and if post curing removes the engineered, varied properties, then that is extremely important to note and quite limiting in the scope/relevance of this work.

Version 1:

Reviewer comments:

Reviewer #1

(Remarks to the Author)

Many improvements to the manuscript were made to address concerns with the data analysis, support for conclusions, and clarity of some parts to demonstrate the multimaterial 3D printing of spatially defined regions of semicrystalline and amorphous polymers from one resin using temperature and light intensity. These improvements include Supplementary Table 2 detailing the polymerization conditions of the polymer objects in this work, explanation of why the previously studied formulation did not achieve property differentiation while newly introduced resin did enable multimaterial 3D printing, depiction of Figure 3A and Figure S16 showing the entire 3D printed structure made with a temperature gradient, and the organized print designs in Figure 5, Figure S25, and Figure S26. However, there still remain a few minor points of concern that should be addressed before considering acceptance:

1) There were some minor adjustments made to the introduction to help with clarity, but some of the highlighted red text is actually not changed from the original version and paragraph 2 on page 3 is still hard to follow. This paragraph reintroduces challenges with greyscale printing again, which was already discussed in the previous paragraph on page 2. The literature examples described on page 3 are not clearly explained with terminology not fully explained (i.e., how “uncatalyzed cycloaddition dimerization reactions” lead to the stiffness tuning even after post-curing, “light-stabilized dynamic materials”, or “property differentiation based on dye oxidation”) and should be compared to the current work. As stated in the Cover Letter, the benefit of the proposed approach is going beyond just “stiff and soft” differentiation in material properties and adjusting these with changing printing temperature, but this is not highlighted clearly in this paragraph. Overall, the introduction is very long with many examples from literature described making it hard to follow what the novelty of this current work is.

2) The added Supplementary Figure 20 is useful for showing print fidelity of the multimaterial properties or how accurately the part matches the design file, but does not accurately show “pixel-to-pixel” property differentiation or resolution, which describes the smallest feature sizes that can be printed. Print resolution from Supplementary Figure 20 is only about 100 μm . It is not possible to have resolution less than the pixel size. The wording “dimensional accuracy of the crystalline-amorphous differentiation” on page 13 is a more appropriate description. The other terms throughout the text should be clarified to describe “resolution” or “dimensional accuracy/fidelity”. For the skeleton print, the dimensional accuracy is described as “maximum deviation of 22 μm ”, but should be clarified what this is deviated from (i.e., the original design) and include the dimensions of the original design in Supplementary Figure 20 for clarity in what the measurements should actually be.

3) In Figure 1, the amounts of liquid crystalline monomer and thiol crosslinker are described in terms of “functional end group percent” or “FG%”, which isn’t a standard unit of measure. A more accurate unit of measure would be 1:1 molar ratio of functional groups or adding the word “mol”, such as “50 mol% of the functional end groups are ‘ene’ end groups from BPLC monomer”.

4) The line separating the sections irradiated at different intensities in the temperature gradient print in Figure 3A and Supplementary Figure 16 is described as green, but I believe it is the gradient line going from black to white.

5) The scale bars were removed from the 3D models in Supplementary Figure 25 and 26. Please add them back for clarity.

Reviewer #2

(Remarks to the Author)

Michael Göschl and co-workers present a substantially revised manuscript describing the fabrication of semi-crystalline and amorphous materials via multi-temperature vat photopolymerization. The authors have addressed all major issues raised in my previous review and the study now meets the scientific and editorial standards of Nature Communications in my opinion, particularly given its implications for multi material additive manufacturing and photonic applications in different research fields.

To further enhance clarity and completeness, I would recommend incorporating the following minor revisions:

1.- Resin stability: The formulation undergoes a marked viscosity increase after ~5 h at 110 °C. While this pot life may suffice for certain prints, longer stability will be required for more complex builds. Please note this limitation explicitly and, where appropriate, mention straightforward mitigation strategies (e.g., incorporation of stabilisers or radical scavengers).

2.- Irradiation intensity versus bulk temperature: High laser intensities can locally elevate resin temperature and may mimic printing at high temperatures as stated by the authors. Could a hybrid approach (moderate bulk temperature combined with local intensity control) accelerate multi material builds by reducing thermal equilibration time? A brief discussion would be valuable.

3.- Commercial availability and future outlook: Because the custom printer and control software are not yet commercially available, the implications for other researchers may be limited. I encourage the authors to acknowledge this constraint and to offer their perspective on future accessibility of “hot lithography” platforms, ideally in the conclusions section.

Besides I think the that Figure 1b would benefit from emphasizing that temperature modulation during printing as the decisive factor that enables amorphous to semi-crystalline domains.

Voxel based design software – Please specify the voxel modelling software (and version) used to generate multi material objects, or provide a concise description if a bespoke tool was employed.

These suggestions are minor and should be readily addressable. I remain enthusiastic about publication of this work in Nature Communications once the above clarifications are incorporated.

Reviewer #3

(Remarks to the Author)

Upon re-review of the manuscript and given the extensive, thoughtful, and thorough follow-up experiments and responses to each reviewer's comments and concerns, I would now recommend the manuscript be accepted without revision.

Reviewer #1

Comment: In this study, multimaterial 3D printed parts using vat polymerization 3D printing is demonstrated to print both semicrystalline and amorphous polymers in spatially defined areas from a single liquid crystalline monomer resin. The amorphous vs. crystalline polymer regions are controlled via printing temperature and light intensity. There are a handful reports in literature related to vat polymerization of semicrystalline polymers, but this study is unique in terms of the use of liquid crystalline monomers for multimaterial printing and the ability to control between amorphous and semicrystalline polymers (with variations in opacity and mechanical properties) with temperature and light intensity in one print and within one layer, as opposed to switching materials during the print. However, there are flaws in the data analysis, some details are missing from the methods, and clarity of the manuscript needs to be significantly improved to better understand the results. The following points require major revision:

Answer: We thank the Reviewer for their positive evaluation of the uniqueness of our presented results in related scientific fields. Further, we are grateful for the detailed insights regarding interpretation and representation of our results, which we have taken to heart and revised as addressed point-by-point below.

Comment: 1. From the bulk and 3D printed results, the importance of both temperature and light intensity in controlling amorphous vs. semicrystalline domains is evident. However, in certain points throughout the text (e.g., the title that mentions only “greyscale printing” and page 6, paragraph 2 mentions “only a simple temperature adjustment”) only one or the other is mentioned. The importance of both should be more clearly stated. Also, the curing/printing conditions were sometimes hard to follow with either only temperature stated or vague light intensity conditions (e.g., “mild irradiation”). Perhaps a summary table would be useful.

Answer: We agree that we should reconsider our use of the terms “greyscale printing” and “multi-temperature printing” and have thoroughly gone through the manuscript, including the mentioned sections, to clarify the different possibilities and limitations to control crystallinity with intensity vs. temperature. While the majority of the experiments describe a “multi-temperature” printing approach, the gradient print in Figure 3A (page 11) demonstrates the possibility for a “greyscale” approach. Additionally, we have proven in a newly added experiment that optical properties can indeed be influenced by irradiation intensity (Supplementary Figure 17, page S19).

To suffice the Reviewer’s request of making the curing/printing conditions throughout the manuscript more transparent for the reader, we have accommodated their suggestion and added a table in the Supplementary Information detailing the curing parameters of all experiments (Supplementary Table 2, page S9).

Comment: 2. The introduction is hard to follow and some arguments for why the proposed technique is a better multimaterial method are conflicting. For instance, it is said that exchanging the material feed is time intensive (page 2, paragraph 1), but changing the temperature and the long exposure times used in this study would also contribute to long print times. The literature examples on page 3, paragraph 1 were difficult to understand. The drawbacks of many other relevant multimaterial 3D printing examples were described in the introduction, but it wasn't always clearly stated how the method in this study is an improvement compared to these.

Answer: Thank you for the feedback regarding our introduction. We have carefully considered all aspects of the Reviewer's suggestions for changes and integrated them in the new version of the manuscript.

Changing materials (which is currently one of very few multi-material SLA/DLP/mSLA approaches) comes with one major drawback: The cleaning process between the materials (which may weaken the material itself and adds print time and complexity). We show in this work that it is also possible to create multi-material prints without this intermediate cleaning step. The long cooling/heating time is mainly because the build platform of the BP10 printer is a massive block of aluminium that is not cooled actively. While this is a current limitation, it can be easily re-engineered either by introducing active cooling or by avoiding the high heat capacity. The vat on the other hand is actively cooled and cools down much faster than the build plate. Therefore, we are confident that our approach allows higher speeds than the current material exchange + cleaning approach. Supplementary Figure 19 (SI page S20) shows the heating/cooling cycle of one multimaterial print layer, which shows that the actively cooled components reach the target temperatures in around 60 seconds. This heating/cooling speed could be reached for all components. Additionally, significant further optimization is possible. We have reflected these thoughts also in the manuscript now. Gradient printing and multi-colour/wavelength printing, the other options for multi-material SLA/DLP/mSLA printing, do not offer the same degree of freedom with respect to property switching as our approach, as already described in the introduction.

Comment: 3. Page 4, paragraph 2 Property differentiation studies – Further support and experimental results should be provided as to why the previous formulation did not achieve property differentiation with printing parameters. Additional support for the choice of the new liquid crystalline ene-monomer (BPLC) should also be provided. Why would an even more highly ordered smectic X phase contribute to property differentiation with printing parameters? Discussion of previous studies (page 4, paragraph 3) is stated, but no reference or data is given.

Answer: Thank you for these important remarks. We have amended this section of the introduction by explaining the limitations of our previously reported liquid crystalline photopolymerizable formulation compared to the one utilized herein: A differentiation between the differently cured materials could not be achieved because the transition of the formulation from liquid crystalline to isotropic liquid occurred only gradually over a broad temperature range. Additionally, the temperature, at which an isotropic liquid state finally occurred, was too high to be achieved within our 3D printing setup. These two aspects made multi-temperature printing unfeasible.

Hence, the current monomer BPLC was tested and found to be better suited to induce property differentiation. The description of the smectic X phase as “even more” highly ordered in this passage may have been misleading, as the high order is not our proposed primary cause of the good differentiation between crystalline and amorphous in our materials. Rather, the sharp transition between the LC and isotropic phase, which occurs even in the presence of a thiol

comonomer impurity is likely the main cause of the difference between crystalline and amorphous. The highly ordered smectic X phase likely provides an additional stimulus toward crystallization when polymerization from the LC phase takes place.

We have adjusted the paragraph to better reflect these aspects. While we do show the supporting data for the choice of the new liquid crystalline ene-monomer (determination of temperature ranges and transitions of LC vs isotropic phases in the monomer formulation via light optical microscopy), we have not added the results of our previous study in this paper again. Instead, we have added citations to our preprint, in which all this data can be found for comparison with the new data reported for BPLC.

Comment: 4. In Figure 2D, please explain the standard deviation lines. How many repeat samples were tested?

Answer: Thank you for noting that we forgot to add the number of samples tested in our methods section. We have now added this to the figure descriptions and to our description of the tensile testing procedure (now in the Supplementary Information Chapter 12, Page S12) and described the standard deviation lines in the caption of Figure 2D and 3D (pages 8 and 11).

Comment: 5. While the bright phenomena above 100 degC was explained in Figure 3A as due to the way the sample was cut and the polarizing filters, it is misleading and makes it appear as if these areas are semicrystalline. Could this be avoided by printing a larger sample and having the cut parts outside of the optical microscope image? Or perform background image correction? Or include other images to prove the bright phenomena position changes with direction of the polarizing filters and isn't part of the sample? Also, the temperature scale is hard to follow (ticks of 8 degC). How many layers were printed at each temperature? Maybe a 3D model of the printed sample with different light intensities and temperatures described could be included to help explain the sample more clearly. There appears to be some bleeding of amorphous region into crystalline region between high and low intensity regions at ~86 degC. Is this real? If so, why?

Answer: Thank you for drawing our attention to the unclarity of Figure 3. We have amended the problems regarding the bright phenomenon above 100 °C by adding proof to the SI, that these are simple imaging artifacts of the polarized microscopy, which cannot be avoided, and hope the new version is now acceptable to the Reviewer. As the entire area of Figure 3A comprises of a microtome-cut surface, unfortunately these bright phenomena are not entirely avoidable and appear in different areas depending on the angle of the polarizer. To prove that the bright sections indeed change their positions depending on polarizer angle, we have added an image in the SI (Supplementary Figure 16, page S19) displaying the full microtome slice, wherein the polarizing filter is rotated slightly, causing different areas of the image to turn bright. Along with this Supplementary Figure we have also elaborated upon the number of layers printed at each temperature and provided an explanation for the bleeding effect (propagation of crystallite nucleation throughout the formulation).

We further agree that it may be difficult to understand Figure 3A without a schematic representation of how it should be understood. We have therefore added more context in the main text (page 9), a more descriptive caption for Figure 3A, and have added the full print conditions and a clearer temperature scale into the graph (printing direction, steps of 10 °C with sub-steps of 2 °C) to make it more clear how the image can be read. Along with Supplementary Figure 16,

which shows the cutout area displayed in the main part, the figure should now be much easier to understand.

Comment: 6. Please provide a reference for the degree of crystallinity equation. The D_c is typically referenced to the 100% crystalline polymer – is it valid to use the BPLC monomer? It is stated that in addition to some bulk and 3D printed samples, a baseline correction could not be found for the BPLC pure monomer and the isotropic enthalpy could not be accurately calculated, yet it was still used as a reference point to calculate D_c for the bulk and 3D printed samples. Please explain how conclusions could be made with this method that could not be properly calculated. Page 6, paragraph 1 – please provide a reference showing that the broad flat peak at 100 degC for the amorphous sample signifies traces of crystallinity. How do these crystallinity values compare to other 3D printed semicrystalline or liquid crystalline polymers, especially in 3D printing?

Answer: The D_c equation was a proposed way to estimate the degree of crystallinity in a situation that did not allow a calculation of the melt enthalpy for a 100% crystalline polymer (which effectively cannot exist, as we always produce semi-crystalline networks). However, we do realize that the formula contains an inherent inaccuracy, as crystallization of any part of the thiol comonomer cannot be not taken into account. Therefore, we have opted to remove the formula for better accuracy and use the ΔH values for comparison. We have compared the obtained ΔH values with a number of sources from literature (main paper page 7), which identify the melt enthalpy of our materials as very high, even when compared to non-crosslinked polymers. We have also added two citations, one of which shows a biphenyl liquid crystalline elastomer at several stages of annealing, in which some of the DSC curves resemble the amorphous polymers (bulk polymerized and printed at 110 °C) in our work, reinforcing the “traces of crystallinity” term (main paper page 6). The second citation also shows DSC curves with a T_g and T_m feature, which diminishes based on the polymer architecture.

Regarding “a baseline correction could not be found for the BPLC pure monomer and the isotropic enthalpy could not be accurately calculated”: A misleading error occurred in the Supplementary Table (now Supplementary Table 3, SI page S10), due to which the LC-isotropic transition was accidentally labeled with the footnote identifier 1, which describes that the baseline could not be calculated. The footnote identifier 1 previously referred to a footnote describing the second transition of the monomer as the LC-isotropic transition. Later, we opted to remove this footnote as we included ΔH_i in brackets in the table header. Thus, we changed the footnote identifier for the samples for which a baseline correction could not be performed from 2 to 1. At this point, we overlooked that the identifier 1 was still present for the LC-isotropic transition. We apologize for this error and thank you for pointing it out. A baseline correction was in fact performed for the sharp LC-isotropic transition. We have removed the erroneous footnote identifier from the ΔH_i transition.

Comment: 7. The claimed “... pixel-to-pixel precision property switching within one printing layer ...” (Abstract, page 1; Introduction, page 3, paragraph 1; page 10, paragraph 3) is not clearly demonstrated, especially when referenced to the QR code in Figure 5A where the smallest feature is ~300 μm .

Answer: We thank the Reviewer for their critical remark regarding visual confirmation of our resolution in the current version of the manuscript and SI. We have now added a figure to the SI, which demonstrates the precise pixel dimensions precisely match the transparent-opaque

transition in our images (Supplementary Chapter 20, Supplementary Figure 20, page S21). Additionally, a brief discussion regarding this aspect has been added. Indeed, the resolution is maximally half-pitch pixel sized!

Comment: 8. Shape memory tests – although addressed with the adjusted shape recovery ratio equation, it is difficult to make the conclusion of outstanding average recovery and fixity when the samples elongate over time and do not contract as is typical. Is this common for liquid crystal polymers? More explanation and comparison to other relevant examples is needed to understand these results.

Answer: Thank you for bringing up this misunderstanding. For our measurement, we used a measurement method that had a difference in programming compared to the literature-known method. Along with the general measurement parameters of strain-controlled programming with stress-free recovery, we always set the elongation to 0% after each cycle to ensure that we always strain the sample to a relative elongation of 50%. We did not stretch to the same maximum length each cycle (as is usually done in literature), but to the same relative elongation of 50%, which appeared more appropriate to us. With this adjustment, the sample elongated over the course of multiple measurement cycles.

Thanks to the Reviewer's comment we have realized that this is not the conventional method usually performed in literature. Therefore, we have performed another measurement using the exact literature-known method (Main part Figure 6A, page 15, SI page S21, Supplementary Table 4, Supplementary Figure 21). The measurement did not result in the upfront excellent performance that was observed in the previous measurements, as the known "training phenomenon" occurred in the first cycle, giving a shape recovery of 80%. After this first cycle, however, again very good shape recoveries were once again achieved. We assume that this discrepancy is due to a slight difference in sample preparation, as the parameters of the first cycle were performed exactly the same way as previous measurements. Nevertheless, we have decided to put the worse-performing, but more in-line with literature measurement in the main paper to avoid confusion, while keeping the data of both methods in the SI.

Comment: 9. The reference to the figures is out of order and/or not in logical locations in the text (e.g., Figure 1A, Figure 2A, Figure 1B, Figure 2B, Figure 3A, Figure 2C, etc.). Also, reference to some Supporting Information calculations, figures, and tables are missing and would be helpful to include in the text to provide further clarification on certain points (e.g., Section 6 Polymer crystallinity calculation, SI Figures 4 – 9, and SI Table 2)

Answer: Thank you for this note, we agree that some additional crossreferences to the SI and a better enumeration sequence could be useful for understanding. We have reworked the main manuscript to reference the figures in the correct order. Additionally, we have reordered the Supplementary Information to strictly reflect the order of experiments described in the main paper and added cross-references to the SI figures and tables.

Comment: 10. Some of the figure captions are not clearly described (e.g., Figure 2 and 3). Separate sentences should be written out for each part of the figure and include more details. Also, "thermomechanical" properties from tensile tests is incorrect if the samples were tested at room temperature (Figure 2 and 3 caption).

Answer: We have amended the requested figure captions as well as further captions according to the suggestion of the Reviewer.

Comment: 11. There is confusion between the five-layer tensile and fiber structure in SI Figure 6 (the fiber tensile test 3D model looks like it is labelled as the five-layer tensile test in the SI). The total 3D model and cross-section models for certain prints should be grouped more clearly together (e.g., hidden QR code, fiber tensile test, five-layer tensile test, etc.) and both in the same document (main document or SI).

Answer: We have revised the order of representation of composite structures as suggested to enhance clarity in Figure 5 and the newly reordered Supplementary Figures 25 and 26 (SI page S26), which show all printed specimens. Of the five-layer structure, we printed full tensile test specimens and performed tensile tests (main part, Figure 3E), while of the fiber structure cross-section, we printed only the cross-section as a 3D-structuring proof of concept. We have now mentioned this in the main part to avoid confusion, and have avoided labeling the fiber structure cross-section as a “tensile test”.

Comment: 12. It is confusing to have some Methods information in the main document and others in the SI or in both (e.g., 3D printing). Choose one or the other. An image of the custom 3D printer and the equipment used to heat vat would be useful. Why were different wavelengths used for the bulk curing and 3D printed samples? Page 14, 3D Printing – “3D printing was performed using stereolithography (SLA) with a digital light processing (DLP) light engine ...” SLA and DLP are two different printing techniques (rastering laser vs. 2D projected light exposure). Please clarify.

Answer: The rationale of having two Methods sections was that novel methods have been described in the Main, while standard methods routinely utilized in the polymer community have been described in the SI. However, we see why this could seem confusing to someone reading the manuscript for the first time and we have now opted to put all information regarding Methods into the SI. We have also added the requested image of the 3D printer in the instruments section of the SI (Supplementary Figure 1, page S4).

Regarding the use of different light sources, we have added context about why different wavelengths were used in the Supplementary Information (SI chapter 9, page S8). We agree that using the same wavelength for all polymer curing would be optimal, however, due to limited equipment availability we were unable to realize this. Therefore, we had to use a 405 nm light source for curing at the low intensity of 1.5 mW cm^{-2} and a 365 nm light source for curing at higher intensity of 290 mW cm^{-2} (which is the minimum intensity of this particular light source). For 3D printing, a specialized printer with accurate temperature control up to relatively high temperatures above $100 \text{ }^{\circ}\text{C}$ was needed, which was only equipped with a 385 nm light source. The comparison of mechanical properties of bulk vs printed specimens, however, demonstrates that equal outcomes at different wavelengths is possible, since the efficiency of the photoinitiator is very good at all these wavelengths.

With respect to the comment regarding printing methods, we agree that there is confusion regarding the terms in the community. We agree that light-based 3D printing would include laser sintering methods as well. SLA in fact refers to the machine (StereoLithographic Apparatus) rather than the technique and yet it has been used frequently to refer to the method of 3D printing where a laser is utilized. Thus, we have now explicitly settled for the use of “DLP printing” only in our manuscript.

Comment: 13. Page 15, paragraph 1 (3D printing method) – Post-processing conditions were different for crystalline and amorphous parts. What post-processing was used for multi-material prints? The amorphous parts had less solvent resistance. Does this mean the part degraded more with the toluene wash for multimaterial parts? Could this have contributed to the change in mechanical properties? Was the difference in transparency between amorphous and crystalline prints similar before and after post-processing in toluene?

Answer: We have expanded the post-processing procedure in the 3D printing methods section to address these questions. This chapter is now located in the Supplementary Information (chapter 14, page S14). We optimized the post-processing procedure with test prints before we post-processed the parts that were later used for mechanical and optical tests to make sure that there is no influence on the test results. The multi-material parts were post-processed the same way as the amorphous parts to prevent swelling of the amorphous sections. We did not notice any difference in transparency before and after any post-processing, even when swelling occurred. This has now been quantified via optical spectroscopy (described in the SI).

Comment: 14. What were the transmission and opacity for samples cured/printed at different light intensities?

Answer: Thank you for pointing out that optical characterization of the printed samples at different intensities of light were missing. We have now supplemented these measurements in the SI (Supplementary Figure 17, page S19). They corroborate our claim that greyscale printing is possible to a certain extent to create the multi-material objects.

Reviewer #2

Comment: Göschl *et al.* report an extension of their previous work on photopolymerizable liquid-crystalline (LC) systems. By modulating printing temperature and light intensity on a newly developed photoresin, the authors transform the same resin into either crystalline or amorphous thermosets, enabling high resolution multi-material printing as well as shape-memory and optically programmable objects.

While the concept is timely and potentially impactful, I find the manuscript's presentation incomplete and, at times, confusing. Critical experimental details, characterization data, and discussions are either missing or insufficiently developed. Consequently, the study does not yet meet the high standard of clarity and impact expected for publication in *Nature Communications*. I therefore recommend rejection in its present form, with the suggestion that the authors consider submitting a revised version to a specialized journal in additive-manufacturing research.

Answer: We thank the reviewer for evaluating our presented concept as timely and potentially impactful. We have taken the Reviewer's concern regarding the manuscript's presentation very seriously and carefully drafted a revised version to meet the high standards of Nature Communications. We appreciate the Reviewer's annotation that the manuscript does not meet the high standard of clarity and impact yet, with the implication that a revised version could. We would also like to emphasize that we do believe that our new multi-material printing approach can indeed be highly valuable beyond a specialized (photo)chemistry and additive-manufacturing audience and therefore is better suited for a multidisciplinary audience as it is the case for the readership of Nature Communications. This conclusion stems from the fact that we do not only demonstrate an approach to print hard/soft materials as other methods have before but also an approach to print highly distinct optical properties from one photoresin. This could be of value to other communities in the physical sciences who are interested in developing novel customizable optical elements, patternable optical filtering and exploring new materials for optically encoded data storage. Furthermore, the implementation of the multi-temperature printing process could be of interest to any engineering-focused audiences as it provides novel mechatronics-oriented solutions, especially regarding the readout of a voxel-based model using layer colour information as well as regarding model systems to understand the effect of microstructures (heterogeneity distribution) in composites on thermomechanical behaviour.

Major Comments

Comment: 1. The Introduction outlines recent advances but falls short of articulating *why* multi-material 3D printing is transformative for additive manufacturing. Please expand this section to contextualize the technological and scientific significance.

Answer: Thank you for challenging our motivation-section in the introduction. We have now expanded it to provide more context, especially along the lines why it is important beyond the additive manufacturing community and photochemistry community itself (also see previous answer).

Comment: 2. The discussion begins with thermal analysis before the resin composition and structural rationale are introduced. Reorder the section so that formulation details preceded characterization results, and revise for better logical flow.

Answer: Thank you for pointing out that the beginning of the chapter needs adjustment. We realize that its previous state did not describe with sufficient clarity, which parts outline previous work, and where the present results begin. We have rewritten the beginning of the “Property Differentiation Studies” chapter to correct this. Additionally, we have reordered the section in which current work is discussed to first include a complete description of the formulation composition, which is followed by the characterization thereof. We have also generally improved the logical flow of the paper by reworking the order of cross-references and reordering the Supplementary Information to better reflect the order of the main manuscript.

Comment: 3. Resin formulation

The choice of the LC monomer is justified, but the role of the trifunctional thiol, photoinitiator concentration, and stabilizer content is not discussed.

Provide in the main text specific information regarding the formulation, including monomer ratios, photoinitiator loading, and any inhibitors used.

Answer: We appreciate the Reviewer’s hint that some of the early considerations regarding resin formulation choices are not clear enough in the current manuscript. With respect to monomer ratios, we require a 1:1 equivalent ratio with respect to thiol and ene endgroups, which does not leave room for variation. We have now added a reference to our previous manuscript regarding the choice of the type of trithiol, since we have conducted a thorough thiol screening there. In brief, the trithiol acts as an impurity to the LC phase and therefore the molecular weight of the trithiol compound should be reduced as much as possible. Multifunctionality is required to achieve networks, as the ene is difunctional. Additionally, as rigid as possible structures are beneficial for the trithiol in order to achieve as stiff networks as possible for the semi-crystalline networks. Additionally, we have added a discussion regarding the choice of photoinitiator and stabilizer type and concentration, which is located on pages 4 and 5 of the main manuscript. An adjustment of the formulation had to be made for the hollow pyramid printing test, which included more layers than the previous prints. More context for this adjustment was added on page 12 of the main manuscript. Additionally, the newly added Supplementary Table 1 provides an overview of the composition of both the formulations used in this work.

Comment: 4. The manuscript lacks kinetic data (e.g., real-time FTIR or photorheology) to quantify monomer conversion and gelation at the different processing temperatures. Besides, I would recommend to include gel content data to ensure high monomer consumption.

Answer: Thank you for pointing out that both should be characterized for the sake of completeness. We have therefore added these experiments now. Photorheology (Supplementary Information chapter 7, Supplementary Figure 6, page S7) demonstrates very fast curing kinetics. Unfortunately, we could only demonstrate this in the liquid-crystalline state of the formulation at 80 °C (measured on the rheometer plate) up to a maximum of 90 °C, since our photorheometer could not be heated to 110 °C without potentially damaging it. Additionally, monomer conversion could not be calculated meaningfully via in-situ collected NIR data during photorheology as the crystallization during curing caused the baseline to shift, preventing a baseline correction. To

compensate for this, we have measured the double bond conversion via ATR-infrared spectroscopy after curing and post-curing (Supplementary Figure 11, SI page S15). The gel content was measured of 3D printed polymers cured at different temperatures and resulted in satisfactory gel fractions over 97% for all curing conditions (Supplementary Information chapter 15, Supplementary Figure 14, page S16).

Comment: 5. The authors state that temperature and light intensity were “adjusted,” but not systematic study is provided in which exposure dose, layer time, or print fidelity is discussed. I recommend including Please including, resin photonic parameters (Jacobs Curve) effect of temperature on resolution objects shrinkage.

Answer: Thank you for pointing out that our method section previously did not describe our systematic way of finding the best printing parameters, even though we did use a systematic approach. This has now been amended (Supplementary Information, page S13). To confirm that our iterative optimization of printing parameters had led to a satisfactory result, we recorded a Jacob’s curve at different curing conditions (Supplementary Figure 10, page S14). We tested the accuracy within the printability window and have shown that the resolution within the multi-material sections of a model is better than half a pixel (25 μm). We observed differences in curing depth depending on the curing conditions. As shown in Supplementary Figure 10, the temperature had a very minor effect on the curing depth, whereas light intensity had a measurable impact. A respective discussion of this finding can also be found in the same section of the SI.

Density measurements of the printed objects revealed that there is no significant difference in the density of crystalline and non-crystalline objects. Comparison to the uncured formulation’s density resulted in a low 2.5 to 3% shrinkage across all samples (Supplementary Figure 15, SI page S18), which should not have a large adverse impact on the printed object’s dimensions.

Comment: 6. Specify the build orientation of tensile/compression specimens relative to the loading axis for test described in Figure 3.

Answer: Thank you for noticing that this information is missing. We have now added it to the Supplementary Information chapter about tensile testing (SI chapter 12, page S12) and shape-memory tests (SI page S22).

Comment: 7. The manuscript mentions adding stabilizer for bulk prints but omits quantitative details. Report stability at different temperatures and any effects on cure rate and final monomer conversion.

Answer: Thank you for pointing out the missing information on stabilizer content, which we have now supplemented (Main manuscript page 5, Supplementary Table 1). We have assessed stability of the most commonly used formulation (all experiments except the hollow pyramid print) via rheology by storing the formulation at the highest processing temperature of 110 $^{\circ}\text{C}$ to determine the stability for the most challenging conditions and found that processing for five hours can be done without any change in viscosity, i.e. onset of polymerization in the absence of light (Supplementary Information chapter 5, Supplementary Figure 3, page S6). By extension, it can be said that the formulation should be stable even longer at lower temperatures, however, the exact

storage time is irrelevant as the curing from the isotropic state at 110 °C is the restricting condition.

Comment: 8. Clarify why the initial proof-of-concept prints used one formulation whereas more complex objects required another. Summarize the compositional differences and their impact on processing.

Answer: The high number of layers led to a much longer printing time, for which a more stable formulation was necessary. Regarding the impact on processing, we opted to use a high-intensity short irradiation program to minimize printing time. We have added details about the formulation composition and reasoning for the adjustment for the complex object print as well as the printing parameters in the main manuscript (page 12).

Comment: 9. Provide dimensional-accuracy (e.g., deviation from CAD) for both crystalline and amorphous regions.

Answer: Thank you for the suggestion, as can be seen in the new figure in the SI (Supplementary Chapter 20, Supplementary Figure 20, page S21), the dimensional accuracy of the different materials exactly exhibits the expected pixel-dimensions with maximum measured deviations of 22 μm (less than half a pixel).

Minor Points

Comment: 1. Figure 2c would benefit from including Tan δ curves to identify glass-transition temperatures unambiguously.

Answer: We have added the full results of the DMA measurements including tan δ curves in the Supplementary Information (Supplementary Figures 8 and 9, Supplementary Information page S11).

Comment: 2. Storage-modulus data is generally plotted in logarithmic scale. IS that the case? Fig. 3 D (for example) shows an abrupt drop of the storage modulus at lower temperatures than T_g , was that expected?

Answer: The storage modulus is often plotted in a logarithmic scale as it may highlight thermal transitions better. However, in our case we opted for a linear scale to better depict the absolute value of G' , which relatively accurately represents the material's stiffness and is therefore preferred in materials science. A logarithmic scale could lead to the G' curve before the T_g being wrongly interpreted as nearly constant, which is not the case for the present materials. The stiffness at different temperatures is an important measure because already the onset of the glass transition determines when significant softening of the material occurs. It is further an important measure for us because it is highly dependent on the sample's crystallinity. Therefore, we believe that a linear scale is better suited for our specific materials.

The beginning of the drop in storage modulus in the DMA graph coincides approximately with the beginning of the increase in slope in the DSC graph. This behavior is indeed typical for the present materials. The glass transition period takes place over a relatively broad temperature

range, e.g., for the amorphous bulk cured sample the $\tan \delta$ peak ranges from 0 to around 55 °C, which is the width of the full glass transition. Around the temperature of the inflection point of the drop in storage modulus, the inflection point also occurs in the DSC graph, which is the characteristic we used to identify the T_g .

One sample (bulk crystalline, mild irradiation, 80 °C), exhibits unusual behavior, as the T_g cannot be identified in the $\tan \delta$ graph, which is why we used the DSC curve to identify the T_g . The likely cause of this is the sample's high crystallinity, which diminishes the impact of the T_g on the material's stiffness.

Comment: 3. In Figure 5a it is difficult to determine printing resolution due to the image scale.

Answer: Thank you for pointing this out. As described in our answer to major comment 9, we have added dimensional accuracy measurements in the SI (Chapter 20, page S21, Supplementary Figure 20) and have also adjusted the zoom-in of Figures 5A and 5B in the main manuscript (page 13) to showcase the pixel-accurate differentiation of material properties better.

Comment: 4. Please include the global printing time per layer on multi-material prints in the same layer.

Answer: We have now included the global printing time for a multi-material layer in our SI (Supplementary Information chapter 19, Supplementary Figure 19, page S20), including precise indications which aspects of the printing process take up how much time. Additionally, we have added information about the printing time of each layer for mono-material prints as well (Supplementary Information chapter 13, page S12)

Reviewer #3

Comment: The authors present a system to fabricate varied mechanical properties using a single vat of material by manipulating the vat temperature during polymerization and, in one experiment, greyscale (grayscale). While the manuscript presents noteworthy results, there are some substantial revisions that would be required for acceptance.

Answer: Thank you for the positive evaluation of our presented concept. In addition to the mentioned varied mechanical properties, we would also like to highlight the varied optical properties achieved by our new approach towards multi-material 3D printing via multi-temperature and greyscale printing. We have taken the Reviewer's comments very seriously and thoroughly included amendments to the manuscript where suggested. Details on this matter are included in the answers to the in-depth comments below.

Comment: 1) The authors used bulk polymerization to determine the mechanical properties, which is known to produce significantly different results than specimen polymerized in a 3D printer. These mechanical properties would need to be repeated using 3D printed specimen (with varied print orientations given properties are known to be dependent on these conditions).

Answer: We agree with the Reviewer that thermomechanical properties may be different in bulk compared to printed specimens. All mechanical tests in Figure 3 were indeed performed on printed specimens as a comparison to the bulk tests displayed in Figure 2. Regarding different orientations, a rotation of the specimen on the printing platform in the horizontal plane would not lead to any change, as each layer is irradiated uniformly. We agree that an upright printing orientation would most likely have a negative impact on the mechanical performance of the printed parts. However, the influence of an upright printing direction in SLA and DLP is well studied and proven to cause a general 15-35% decrease in tensile strength (Li Y, Teng Z. Effect of printing orientation on mechanical properties of SLA 3D-printed photopolymer. *Fatigue Fract Eng Mater Struct.* 2024; 47(5): 1531-1545. doi:[10.1111/ffe.14265](https://doi.org/10.1111/ffe.14265), now also cited in the paper). As we described in the section about printing the hollow pyramid specimen, it was already necessary to enhance the stability of our formulation to be able to print a 7 mm (140 layers) high printed specimen. This, in turn, did not allow for the level of differentiation of crystalline and amorphous properties demonstrated in Figures 2 and 3. Printed upright mechanical test specimens would have a height of 35-40 mm (up to 800 layers), which, even if the printing process could be made possible through a drastically higher stabilizer content, would most likely not result in the desired property differentiation. We have added information in which orientations the parts were printed in the latter part of the 3D printing chapter in the Supplementary Information (SI chapter 13, page S14) along with this reasoning. Information in which orientation the tensile tests were performed was added to the tensile testing procedure (SI chapter 12, page S12).

Comment: 2) Given the title of the manuscript explicitly references the use of greyscale printing (in the pdf, which is different than the title listed here), more thorough investigation into the effects of printing these structures under varied intensities is required in addition to the thermal work. Given the simplicity of changing the greyscale in a given projected layer, this seems like it would have been the first, simplest consideration, which leads me to infer that the authors encountered issues with these efforts that are not adequately disclosed/discussed.

Answer: Thank you for pointing out the difference between titles, which we have amended now. Indeed, the Reviewer correctly assumes that the simpler intensity change during printing does bear a disadvantage, which we can amend by multi-temperature printing: In our current proof of principle, the polymerization heat was able to lower the degree of crystallinity significantly. However, it would require further tweaking of the system to achieve fully amorphous structures just via polymerization heat, if the overall temperature of the vat is not changed, as it is rather energy intensive to destroy the order of the liquid crystalline phase locally. Hence, we found that the additional use of temperature-switching gave a more optimized property differentiation. We have hence amended the title to use the more suitable term “multi-temperature printing” as intended already for the original version of our manuscript. We have additionally extended our discussion, in how far the greyscale printing approach alone works to differentiate the properties (Main manuscript page 9) and added an experiment in which light intensity is varied at a constant temperature (Supplementary Figure 17, page S19).

Comment: 3) Given the breadth of novel chemistries being developed to produce a range of mechanical properties from a single 'pot' of resin in a given print, both a more thorough, recent literature review of these efforts is required. Also, further demonstration of the utility of this technique/approach is required.

Answer: Thank you for critically evaluating our literature review. We agree that we could be even more extensive on the research regarding multi-material printing of varying mechanical properties and have therefore enhanced this aspect in our introduction. At the same time, we would like to emphasize that we do not only change the thermomechanical properties of the material but also the optical properties, a currently underexplored property differentiation in multi-material printing.

Regarding the requested further demonstration of the utility of this technique/approach, we are unsure what the Reviewer refers to. We provide extensive experimental evidence that new data storage and obscuration techniques are possible with this approach, shape memory of the material as well as ample thermomechanical characterization. We would be happy to oblige to the comment upon further clarification what aspects are currently missing/underrepresented.

Comment: 4) It is concerning that the authors did not conduct an additional post curing steps after polymerization given the photoinitiator used is known to be a photoabsorber which would produce a gradient in conversion (thus properties) both within and, likely, between the printed layers. The only remedy for this is to introduce an additional curing step after the part has all excess monomer removed. More explanation as to why the post-curing step was not conducted is required and if post curing removes the engineered, varied properties, then that is extremely important to note and quite limiting in the scope/relevance of this work.

Answer: We have disregarded post-curing due to the high initial double bond conversion directly after polymerization. A gel fraction measurement without post-curing, which we have now included in the SI, resulted in 97% gel content or higher for all samples (Supplementary Figure 14, SI page S17). To thoroughly investigate any possible impact of post-curing on our materials, we have performed analysis of reported properties before and after post-curing: double bond conversion via ATR-IR-spectroscopy (Supplementary Figure 11, SI page S15), optical properties via microscopy (Supplementary Figure 12, SI page S16), thermal analysis via DSC (Supplementary Figure 13, SI page S16), optical properties via IR/Vis spectroscopy (Supplementary Figure 17, page

S19) and are happy to report that no significant differences could be found due to post-curing. This is very much in line with our expectations due to the initially very high double bond conversions.

Reviewer #1

Comment: Many improvements to the manuscript were made to address concerns with the data analysis, support for conclusions, and clarity of some parts to demonstrate the multimaterial 3D printing of spatially defined regions of semicrystalline and amorphous polymers from one resin using temperature and light intensity. These improvements include Supplementary Table 2 detailing the polymerization conditions of the polymer objects in this work, explanation of why the previously studied formulation did not achieve property differentiation while newly introduced resin did enable multimaterial 3D printing, depiction of Figure 3A and Figure S16 showing the entire 3D printed structure made with a temperature gradient, and the organized print designs in Figure 5, Figure S25, and Figure S26. However, there still remain a few minor points of concern that should be addressed before considering acceptance:

— **Answer:** Thank you for acknowledging our thoroughness with the revisions. We have now addressed the final points of concern in the same way.

— **Comment:** 1. There were some minor adjustments made to the introduction to help with clarity, but some of the highlighted red text is actually not changed from the original version and paragraph 2 on page 3 is still hard to follow. This paragraph reintroduces challenges with greyscale printing again, which was already discussed in the previous paragraph on page 2. The literature examples described on page 3 are not clearly explained with terminology not fully explained (i.e., how “uncatalyzed cycloaddition dimerization reactions” lead to the stiffness tuning even after post-curing, “light-stabilized dynamic materials”, or “property differentiation based on dye oxidation”) and should be compared to the current work. As stated in the Cover Letter, the benefit of the proposed approach is going beyond just “stiff and soft” differentiation in material properties and adjusting these with changing printing temperature, but this is not highlighted clearly in this paragraph. Overall, the introduction is very long with many examples from literature described making it hard to follow what the novelty of this current work is.

Answer: Thank you for the critical evaluation of the Introduction. Indeed, we acknowledge all efforts made in the 3D printing community towards multi-material 3D printed objects that have prepared the way towards our newly introduced multi-temperature printing method. While we understand that this makes the introduction longer, we would like to keep all references to the existing literature as we find it important to acknowledge all efforts in the field, without bias, thereby enabling readers to decide for themselves, which methods are relevant to them.

We further thank the Reviewer for clarification, which examples have previously been mentioned in more than one section of the introduction underexplained in the introduction and hope to have amended this shortcoming satisfactorily in the final version through more thorough explanations:

1. The two instances where greyscale printing is mentioned in the introduction were unified in one paragraph:

“Therefore, more recent multi-material vat photopolymerization strategies have focused on the change of material properties through the change in irradiation conditions, light intensity and light colour.² In greyscale printing (in analogy to a black-and-white printer), the intensity of incident light is varied to alter the conversion of the resin into the polymer network and thereby alter its thermomechanical behaviour. Hard and soft sections within one object have been fabricated from

(meth)acrylate resins at high and low irradiation intensities, respectively, by changing the conversion.^{6,7,8} However, while significant adjustments in curing parameters and therefore monomer conversion can induce moderate property changes, the fundamental type of material (stiff vs. soft, tough vs. brittle) is determined by the type and composition of the building blocks.^{9,10}

2. In analogy to this example, some other instances of double-mentioning have been eliminated in the paragraph focusing on the type of material differentiation:

“Most greyscale and multi-color/wavelength printing approaches limit the property differentiation to stiff/soft,^{16,17,18,19,20,21,22} although the differentiation of some other property combinations has been investigated more recently as described above. Additionally, there is one limitation common to all these approaches: [...]”

3. The mentioned examples were explained in more detail:

“Another property differentiation method affects the colour of 3D printed samples via greyscale printing, and is based on the oxidation state of a dye as additive: At high radical concentrations due to high irradiation intensities the dye is oxidized during printing, opposite non-oxidized dye at low irradiation intensity.¹¹ Similarly, pH-changes have been exploited to change the colour of additives in the printing formulation.¹² Again, post-curing could hamper the colour differentiation in the long term. Besides free radical photopolymerization, light-triggered cycloaddition reactions such as [2+2] dimerization reactions of chromophores pendant to prepolymer chains were utilized as crosslinking reactions in greyscale printing. This led to high-resolution stiffness tuning without decreasing the property differentiation via post-curing.¹³ This effect is based on the drastically decreased probability of bond formation or dissociation post-printing because the unreacted groups are immobilized on prepolymers in a rigid matrix, making dimerization in the printed bulk materials highly unlikely. In a variation of this approach, a different type of photocycloaddition dimerization was utilized as crosslinking reaction for prepolymers. Herein, cycloaddition adducts are produced under green light that are unstable if the forward reaction is not continuously triggered by green light.¹⁴ The formed crosslinked materials thus degrade in the absence of light. Tuning the crosslinking density via greyscale printing tremendously influences the likelihood of cycloadduct reversion and hence ability of the materials to degrade in darkness: Above a certain crosslinking threshold at high laser energies, entirely undegradable objects were produced.“

“Most frequently, printing of hard and soft material sections in one object is reported for multi-colour printing.¹⁹ This technique relies on semi-orthogonal photopolymerization reactions, where a radical and a cationic photopolymerization reaction can be triggered in two different wavelength-regimes (ultraviolet (UV) and visible (vis) light), thereby printing different polymer networks when irradiated with UV or vis light. Importantly, these reactions are not mutually exclusive, because UV light typically also triggers the reaction intended for the vis light range, hence the term semiorthogonal. This combination allows for selective vis-induced radical polymerization of a soft (meth)acrylate network in the presence of a UV-active and vis-inactive photoacid generator for cationic photopolymerization. In the UV-region, however, both photoinitiators trigger network formation and a much stiffer, interpenetrating network is created.^{20,21} Further studies have added post-modification steps to this approach to vary the dangling chain ends in the soft network and thereby material properties.²² Another recent report also demonstrates degradable/non-degradable material properties from one resin. Herein, as already introduced previously, two chromophore-based prepolymers are mixed and crosslinked via light-triggered cycloaddition dimerization reactions under conditions that allow fully orthogonal deposition of either polymer network onto the printing platform without the other.²³

Since only one of the two crosslinking reactions is reversible under UV light, this material can be degraded selectively while the second material remains unobstructed.”

4. We also appreciate the hint that we do not highlight the novelty of our approach sufficiently. We believe that the following sections do highlight the novelty of our method:

“Therefore, we suggest a paradigm change in multi-material printing by rethinking the approach of obtaining varying material properties from a selection of monomers. Instead of ascribing different material properties to different monomers, we envision “switchable monomers”, which change their functionality based on the printing parameters, thereby affecting the obtained material properties.

Herein, we propose the printing temperature as a parameter to alter the material properties mid-printing. We suggest our recently established liquid crystalline thiol-ene monomer platform²⁴ as a means to affect crystallinity of resulting polymer networks via the printing temperature instead of previously utilized property tuning via alignment through rubbed polymer surfaces, stretching of prepolymers, or external fields.^{25,26”}

5. However, we agree with the Reviewer that we do not highlight the novelty, that different material properties (transparent vs opaque) are investigated compared to previous work (stiff vs soft) in the current version. We have therefore added a sentence to the previously cited section, to highlight also this aspect of our work:

“This goes beyond previously reported multi-property printing of stiff vs. soft materials, as we additionally focus on optical property differentiation.”

Comment: 2. The added Supplementary Figure 20 is useful for showing print fidelity of the multimaterial properties or how accurately the part matches the design file, but does not accurately show “pixel-to-pixel” property differentiation or resolution, which describes the smallest feature sizes that can be printed. Print resolution from Supplementary Figure 20 is only about 100 μm . It is not possible to have resolution less than the pixel size. The wording “dimensional accuracy of the crystalline-amorphous differentiation” on page 13 is a more appropriate description. The other terms throughout the text should be clarified to describe “resolution” or “dimensional accuracy/fidelity”. For the skeleton print, the dimensional accuracy is described as “maximum deviation of 22 μm ”, but should be clarified what this is deviated from (i.e., the original design) and include the dimensions of the original design in Supplementary Figure 20 for clarity in what the measurements should actually be.

Answer:

Print resolution of the skeleton model in Supplementary Figure 20 (Now Supplementary Discussion Figure 18, SI page S18) is indeed 50 μm (each printed voxel has a size of $50 \times 50 \times 50 \mu\text{m}^3$). We have now added a zoomed in section of the voxel model in Supplementary Discussion Figure 18 for a more accurate comparison. We believe that the single-pixel width features of the mouth of the skeleton are sufficient proof of a pixel-to-pixel differentiation of crystalline and amorphous properties.

Thank you for diverting our focus to the unclear wording in the main text, we have now specified that the deviation of 22 μm is a deviation from the original design (Main manuscript page 13).

Comment: 3. In Figure 1, the amounts of liquid crystalline monomer and thiol crosslinker are described in terms of “functional end group percent” or “FG%”, which isn’t a standard unit of measure. A more accurate unit of measure would be 1:1 molar ratio of functional groups or adding the word “mol”, such as “50 mol% of the functional end groups are ‘ene’ end groups from BPLC monomer”.

Answer: For enhanced clarity, we have changed the unit to mol% end groups in the figure and the manuscript. Main manuscript Figure 1, Page 6.

Comment: 4. The line separating the sections irradiated at different intensities in the temperature gradient print in Figure 3A and Supplementary Figure 16 is described as green, but I believe it is the gradient line going from black to white.

Answer: Thank you for pointing out the error in the description. We have now adjusted the description to reflect the correct colour of the line. Supplementary Discussion Figure 14, Supplementary Information Page S15.

Comment: 5. The scale bars were removed from the 3D models in Supplementary Figure 25 and 26. Please add them back for clarity.

Answer: Thank you for pointing out that the scale bars were left out during the process of rearranging the images. We have added them back into the figures, which are now found in Supplementary Figures 3 and 4 on Supplementary Information page S3.

Reviewer #2

Comment: Michael Göschl and co-workers present a substantially revised manuscript describing the fabrication of semi-crystalline and amorphous materials via multi-temperature vat photopolymerization. The authors have addressed all major issues raised in my previous review and the study now meets the scientific and editorial standards of Nature Communications in my opinion, particularly given its implications for multi material additive manufacturing and photonic applications in different research fields.

To further enhance clarity and completeness, I would recommend incorporating the following minor revisions:

Answer: We welcome the Reviewer's recommendation and enthusiasm for our work and have addressed all suggested revisions.

Comment: 1. Resin stability: The formulation undergoes a marked viscosity increase after ~5 h at 110 °C. While this pot life may suffice for certain prints, longer stability will be required for more complex builds. Please note this limitation explicitly and, where appropriate, mention straightforward mitigation strategies (e.g., incorporation of stabilisers or radical scavengers).

Answer: We have now amended the passage in the main manuscript with an explanation that for even longer printing experiments, further stabilization may be necessary (which was indeed utilized for the hollow pyramid printing test, in which a higher stabilizer content was used). Main Manuscript page 5.

Comment: 2. Irradiation intensity versus bulk temperature: High laser intensities can locally elevate resin temperature and may mimic printing at high temperatures as stated by the authors. Could a hybrid approach (moderate bulk temperature combined with local intensity control) accelerate multi material builds by reducing thermal equilibration time? A brief discussion would be valuable.

Answer: Indeed, this is a highly interesting question, which we have of course considered. However, we have opted to not include this discussion, since it would enter a highly hypothetical realm rather suitable for a perspective than a research article. Many unknown factors such as heat dissipation in the resin make this suggestion far too speculative at this point in time. We believe that our approach to utilize the reaction heat for heating demonstrates sufficiently, that laser intensity is important and would not like to comment on this matter beyond this demonstration.

Comment: 3. Commercial availability and future outlook: Because the custom printer and control software are not yet commercially available, the implications for other researchers may be limited. I encourage the authors to acknowledge this constraint and to offer their perspective on future accessibility of "hot lithography" platforms, ideally in the conclusions section.

Answer: Indeed, the utilized printer in this work is custom-made and not commercially available. However, the underlying technology of Hot Lithography in general is patented and commercially available through the company Cubicure. Furthermore, the implementation of this technique may also be feasible for the adaptation of existing research-prototype machines for DLP 3D printing.

Minor comment 1: Besides I think the that Figure 1b would benefit from emphasizing that temperature modulation during printing as the decisive factor that enables amorphous to semi-crystalline domains.

Answer: Thank you for the comment. We have added a paragraph in the main manuscript and the description of Figure 1 to clarify that the temperature setting is the main parameter through which the crystallinity of the printed part is influenced. Main Manuscript pages 5,6, Figure 1 caption.

Minor comment 2: Voxel based design software – Please specify the voxel modelling software (and version) used to generate multi material objects, or provide a concise description if a bespoke tool was employed.

Answer: Multi-material objects were created in Microsoft Paint – each image corresponding to one printed layer. For the 3D visualization, we employed a voxel-based software that is derived from a Master’s thesis project (<https://repositum.tuwien.at/handle/20.500.12708/20203>). As this software was used solely for visualization and is still in a developmental state, we decided to not release it publicly. The code for the readout of the multimaterial images has been published on TU Wien Research Data and is accessible there. We have added a Code Availability Statement as suggested by the Editorial Office. We have added a passage in the Methods section in the main manuscript describing these details (page 18).

Reviewer #3

Comment: Upon re-review of the manuscript and given the extensive, thoughtful, and thorough follow-up experiments and responses to each reviewer's comments and concerns, I would now recommend the manuscript be accepted without revision.

Answer: Thank you very much for acknowledging of our thorough revisions.

Review Summary:

Göschl *et al.* report an extension of their previous work on photopolymerizable liquid-crystalline (LC) systems. By modulating printing temperature and light intensity on a newly developed photoresin, the authors transform the same resin into either crystalline or amorphous thermosets, enabling high resolution multi-material printing as well as shape-memory and optically programmable objects.

While the concept is timely and potentially impactful, I find the manuscript's presentation incomplete and, at times, confusing. Critical experimental details, characterization data, and discussions are either missing or insufficiently developed. Consequently, the study does not yet meet the high standard of clarity and impact expected for publication in *Nature Communications*. I therefore recommend rejection in its present form, with the suggestion that the authors consider submitting a revised version to a specialized journal in additive-manufacturing research.

Major Comments

1. The Introduction outlines recent advances but falls short of articulating *why* multi-material 3D printing is transformative for additive manufacturing. Please expand this section to contextualize the technological and scientific significance.
2. The discussion begins with thermal analysis before the resin composition and structural rationale are introduced. Reorder the section so that formulation details preceded characterization results, and revise for better logical flow.
3. Resin formulation
The choice of the LC monomer is justified, but the role of the trifunctional thiol, photoinitiator concentration, and stabilizer content is not discussed.
Provide in the main text specific information regarding the formulation, including monomer ratios, photoinitiator loading, and any inhibitors used.
4. The manuscript lacks kinetic data (e.g., real-time FTIR or photorheology) to quantify monomer conversion and gelation at the different processing temperatures. Besides, I would recommend to include gel content data to ensure high monomer consumption.
5. The authors state that temperature and light intensity were "adjusted," but not systematic study is provided in which exposure dose, layer time, or print fidelity is discussed. I recommend including Please including, resin photonic parameters (Jacobs Curve) effect of temperature on resolution objects shrinkage.
6. Specify the build orientation of tensile/compression specimens relative to the loading axis for test described in Figure 3.
7. The manuscript mentions adding stabilizer for bulk prints but omits quantitative details. Report stability at different temperatures and any effects on cure rate and final monomer conversion.

8. Clarify why the initial proof-of-concept prints used one formulation whereas more complex objects required another. Summarize the compositional differences and their impact on processing.
9. Provide dimensional-accuracy (e.g., deviation from CAD) for both crystalline and amorphous regions.

Minor Points

1. Figure 2c would benefit from including $\tan \delta$ curves to identify glass-transition temperatures unambiguously.
2. Storage-modulus data is generally plotted in logarithmic scale. IS that the case? Fig. 3 D (for example) shows an abrupt drop of the storage modulus at lower temperatures than T_g , was that expected?
3. In Figure 5a it is difficult to determine printing resolution due to the image scale.
4. Please include the global printing time per layer on multi-material prints in the same layer.